



# High-resolution debris cover mapping using UAV-derived thermal imagery: limits and opportunities

Deniz Tobias Gök[1], Dirk Scherler[1,2], Leif Stefan Anderson[1,3]

[1]GFZ German Research Centre for Geosciences, Telegrafenberg, 14473 Potsdam, Germany
[2]Institute of Geological Sciences, Freie Universität Berlin, 12249 Berlin, Germany
[3]University of Utah, Salt Lake City, 000, United States

*Correspondence to*: Deniz Tobias Gök (d_goek@gfz-potsdam.de)

**Abstract.** Debris-covered glaciers are widespread in high mountain ranges on Earth. However, the dynamic evolution of debris-covered glacier surfaces is not well understood, in part due to difficulties of mapping debris cover thickness in high
spatiotemporal resolution. In this study we present land surface temperatures (LST) and its diurnal variability measured from an unpiloted aerial vehicle (UAV) at high spatial resolution. We test two common approaches to derive debris thickness maps by (1) solving a surface energy balance model (SEBM) in conjunction with meteorological reanalysis data and (2) least squares regression of a rational curve using debris thickness field measurements. In addition, we take advantage of the measured diurnal temperature cycle and estimate the rate of change of heat storage within the debris cover. Both approaches resulted in
debris thickness estimates with a RMSE of 6 to 8 cm between observed and modelled debris thicknesses, depending on the time of the day. The diurnal variability of the LST controls the relationship between LST and debris thickness and the non-linearity increases with increasing LST. During the warming phase of the debris cover, the LST depends strongly on the terrain aspect, rendering clear-sky morning flights that do not account for aspect-effects problematic. Our sensitivity analysis of various parameters in the SEBM highlights the relevance of the effective thermal conductivity when LST is high. Residual
and variable bias of UAV-derived LSTs during a flight require calibration, which we achieve with bare ice surfaces. The model performance would benefit from more accurate LST measurements, which are difficult to achieve with uncooled sensors.

## 1   Introduction

Debris-covered glaciers are common in many mountain ranges globally. Although debris cover is generally rather thin, usually less than a meter, it has a profound influence on surface melt rates, and thus the mass balance of glaciers. Whereas thin debris
cover ($< 2$ cm) accelerates melt rates, due to the lower albedo compared to clean ice, thick debris cover insulates the ice surface and reduces melt rates (e.g., Østrem, 1959; Nicholson and Benn, 2006). Consequently, heavily debris-covered glaciers can persist longer at lower elevations than debris-free glaciers (Scherler et al., 2011a). Debris-free glaciers worldwide respond to climate change by thinning and retreating (Bolch et al., 2012; Hock et al., 2019; Hock and Huss, 2021). Debris-covered glaciers in contrast show a broad range of responses to climate change as some glaciers are advancing, others are stationary, and some
are retreating (Scherler et al., 2011b; Benn et al., 2012; Gardelle et al., 2012; Kirkbride and Deline, 2013; Benn and Evans, 2014). Therefore, regional to global scale predictions of glacier evolution in response to climate change need to account for debris cover (Rounce and McKinney, 2013; Pellicciotti et al., 2015).

Complex interactions between the various elements of debris-covered glaciers (Benn et al., 2012; Anderson et al., 2021), including differential melt, the overall downglacier thickening of the debris layer, and the presence of ice cliffs and surface
ponds (Kirkbride, 1993; Irvine-Fynn et al., 2017; Miles et al., 2018; Anderson and Anderson, 2018) remain not well understood. Processes responsible for the extent and thickness of debris cover are the rate of debris supply from bedrock hillslopes, the rate of ablation, which exposes englacially transported debris, and surface processes as well as ice dynamics. As all these processes vary with time, supraglacial debris cover changes in time, too. Indeed, recent studies document changes in debris cover thickness in various mountain ranges on Earth. Most studies, however, focus on changes in the extent of debris



cover  (Shukla et al., 2009; Bhambri et al., 2011; Glasser et al., 2016; Gibson et al., 2017; Tielidze et al., 2020), whereas studies documenting changes in thickness are relatively rare (Stewart et al., 2021). In addition, debris thickness observations based on satellite imagery are at best limited to a relatively coarse spatial resolution of tens of meters. In particular, surface processes, such as supraglacial streams, the growth and decay of ponds and cliffs are expected to vary rapidly across the glacier surface (Anderson et al., 2021). A better understanding of transport and emergence of supraglacial debris over short timescales requires

the development of quantitative models. Therefore, comprehensive observations of debris-cover extent, thickness and distribution at high resolution are essential for understanding the dynamic evolution of debris-covered glacier surfaces.

Existing approaches to spatially quantify debris thickness comprise (1) the extrapolation of point or cross section field data (McCarthy et al., 2017; Nicholson and Mertes, 2017), (2) the exploitation of the relationship between the land surface temperature (LST) and debris thickness (Østrem, 1959; Nakawo and Young, 1981), (3) the estimation of melt by DEM

differencing and inversion of the Østrem-curve (Rounce et al., 2018), (4) a combination of 2 and 3 (Rounce et al., 2021) and (4) the use of synthetic-aperture radar (Huang et al., 2017). It has been shown that the LST can be related to debris thickness by extrapolating empirical functions (e.g. linear, exponential, rational) using ground data (Mihalcea et al., 2008; McCarthy, 2019; Boxall et al., 2021), exponential scaling assuming the lowest measured LST corresponds to 1 cm debris thickness (Kraaijenbrink et al., 2017), or solving a surface energy balance model for debris thickness with meteorological data input

from either automated weather stations or reanalysis data (Zhang et al., 2011; Foster et al., 2012; Rounce and McKinney, 2013; Schauwecker et al., 2015; Stewart et al., 2021).

Recent technological advancements allow for the estimation of LST at high spatial resolutions on glaciers. LST can be measured in high resolution using uncooled microbolometers applied either obliquely from the ground surface (Hopkinson et al., 2010; Aubry-Wake et al., 2015; Aubry-Wake et al., 2018) or in nadir mounted to an unpiloted aerial vehicle (UAV)

(Kraaijenbrink et al., 2018). Both applications to measure LST hold various opportunities and limits.  Debris thickness was recently mapped using oblique LST (Herreid, 2021), but the quantification of debris thickness from UAVs has remained elusive. The possibility to measure the spatiotemporal variability of LST from ground or UAV is a particular advantage, as most thermal infrared measurements from space do not have a sub-daily temporal resolution.

Here, we present UAV-derived LST's and its diurnal variability to estimate debris thickness as it varies in space for various

times of a day. To estimate debris thickness we, solve a surface energy balance model using ERA-5 reanalysis data and the measured LSTs. We take advantage of the diurnal measurements and also consider the change in heat storage of the debris as part of the surface energy balance model. We then compare the results with debris thickness maps derived from the empirical relationship of LST and in-situ measured debris thicknesses using a rational curve.

## 2    Study Area

70        Tsijiore-Nouve Glacier (TNG) (Fig. 1) located in southwest Switzerland (46.01, 7.46) is around 5 km long with an average width of ~300 m. The surface area of TNG covers ~2.73 km². The glacier is characterized by an ice fall in the central part, separating the debris-covered and the debris-free part of the surface. The flow direction is north and shows a strong eastward knickpoint within the ablation zone. The lateral moraines are very steep and partly vegetated. The surface of TNG hosts steep ice cliffs (Fig. 1b), supraglacial streams, debris-free bare ice parts, partly continuous as well as partly patchy debris-

cover of heterogenic thicknesses and grain sizes. The glacier is easily accessible at day and night and therefore well suited for our study. The study focuses on a nearly continuously debris-covered portion of TNG. A relatively small study area allowed for numerous UAV flights covering the entire study area throughout the day.


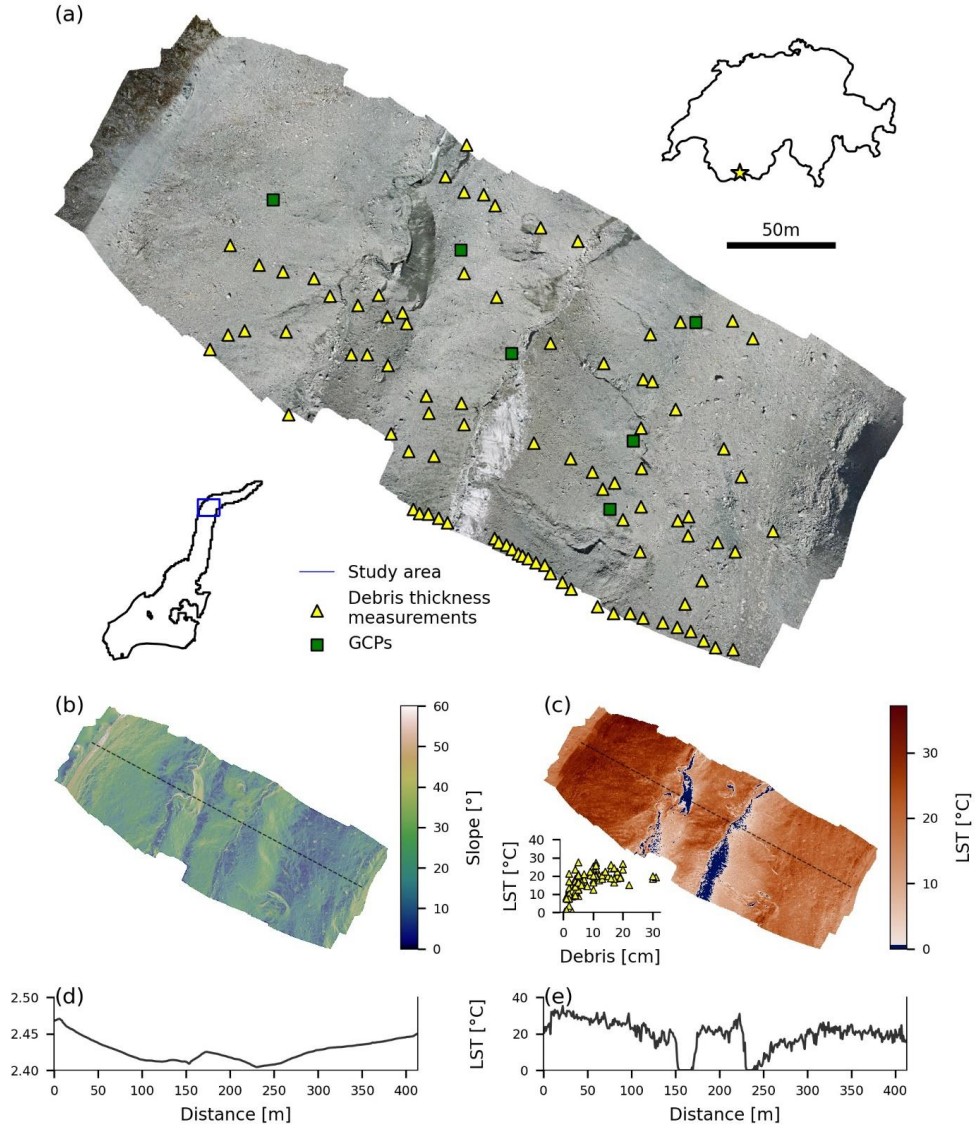

**Fig. 1.** Overview of study area on Tsijiore-Nouve Glacier, Switzerland. (a) Orthomosaic from optical unpiloted aerial vehicle (UAV) data obtained on 30.08.2019. Yellow triangles indicate locations of debris thickness measurements, and green squares indicate ground control points. (b) Slope map obtained from UAV-derived digital elevation model (15 cm resolution). (c) UAV-derived land surface temperature (LST) at 13h. Blue areas depict LST < 0.5 °C. Inset scatter plot shows in-situ debris thickness measurements versus LST. Black dashed lines in (b) and (c) indicate profiles shown in (d) (elevation) and (e) (LST).

## 3    Materials & Methods

### 3.1    Field Data

Field data were collected on 30.08.2019 on an area of approximately 60000 m$^2$ (Fig. 1a). The meteorological conditions were mostly under blue sky conditions (isolated clouds in the late afternoon). Spatially distributed debris surface temperature was measured between 9 h and 22 h at 2-h intervals to capture the diurnal temperature cycle. Temperature measurements were done using a radiometric uncooled microbolometer (FLIR Tau2 longwave infrared thermal camera) mounted to a DJI Mavic Pro UAV. The UAV followed the same pre-define path for all 8 flights at 80 m elevation above the





glacier surface (terrain adjusted). Optical UAV imagery (12 MP) was recorded simultaneously with the thermal images. The thermal sensor operates within a temperature range of -40 °C to 160 °C, has a resolution of 640x512 pixels and measures longwave radiation within a range of 7.5 to 13.5 μm. Recording of the thermal infrared images was done in conjunction with the ThermalCapture 2.0 OEM (Teax Technology GmbH), allowing for the storage of images on an SD card. Recording was

done with a reduced framerate (default 9 Hz). The set-up is suitable for high mountain UAV applications due to the very low size and weight. Prior to the UAV flights 6 ground control points (GCPs) were distributed across the area of interest (Fig. 1a). The GCPs are made from aluminium foil what makes them clearly recognizable in the thermal images due to the very low emissivity.

Debris thickness measurements were made at 90 locations within the study area (Fig. 1). Coordinates of measurement

locations were documented using a Garmin Handheld GPS device (horizontal accuracy: ±3.6 m). The debris cover on the TNG is generally thin: measured thicknesses are below 30 cm, with a mean of 9 cm and a standard deviation of 10 cm. The debris cover close to lateral moraines consists of very large boulders (>0.5 m) that rendered measurements impractical. Furthermore, it was not entirely clear where debris-covered ice transitioned to lateral moraines near the glacier margin.

### 3.2 Thermal image correction

Uncooled microbolometers are sensitive to environmental temperature fluctuations (Heinemann et al., 2020). Specifically, the sensor's detector focal plane array, the sensor housing, and the lens of uncooled microbolometers are sensitive to temperature changes. Accurate radiometric temperature measurements require a thermal equilibrium between the sensor's components and the environment. Unbalanced thermal conditions introduce a temperature bias. The thermal adjustment of the sensor can thus lead to changes in measurements, known as thermal drift: the recorded temperature changes while the object's

temperature remains the same (Ribeiro-Gomes et al., 2017; Malbéteau et al., 2018; Dugdale et al., 2019a; Aragon et al., 2020). Furthermore, the ever-changing micro-meteorological conditions under a drone prevent the perpetuation of a thermal equilibrium and hamper accurate radiometric measurements.

The FLIR Tau2 sensor performs an internal calibration, the flat field correction (FFC), to correct for non-uniformities by lens distortions and variations in the thermal pixel-to-pixel sensitivity. FFC is performed using the shutter at power up,

when the camera changes temperature, and periodically during operation. The shutter is considered to be a uniform temperature source for each pixel and is used to update the offset correction coefficients. This internal calibration leads to in-flight temperature jumps that are accounted for in a post-processing step, called drift compensation. The occurrence of the FFC events is used to calculate linearly backward an offset-value for each frame (Teax, personal communication). Usually this is done automatically by the ThermalViewer software but in our case, the reduced framerate resulted in the loss of several frames

containing the FFC occurrence metadata entry. A drawback of the system one should be aware of. However, we identified the frames following the internal calibration and implemented the drift compensation ourselves (Fig. 2b). To find the temperature jumps within the images, we used a threshold of 2 K differences in the mean temperature of the overlapping part in consecutive image pairs. The temperature jumps are clearly visible in the histogram time series (Fig. 2a) and we found this threshold to match the temperature jumps best. The overlap was defined as the bounding box of matching keypoints detected in successive

images using the Oriented FAST and Rotated BRIEF (ORB) algorithm (Rublee et al., 2011) implemented in the scikit-image python library (Van Der Walt et al., 2014).





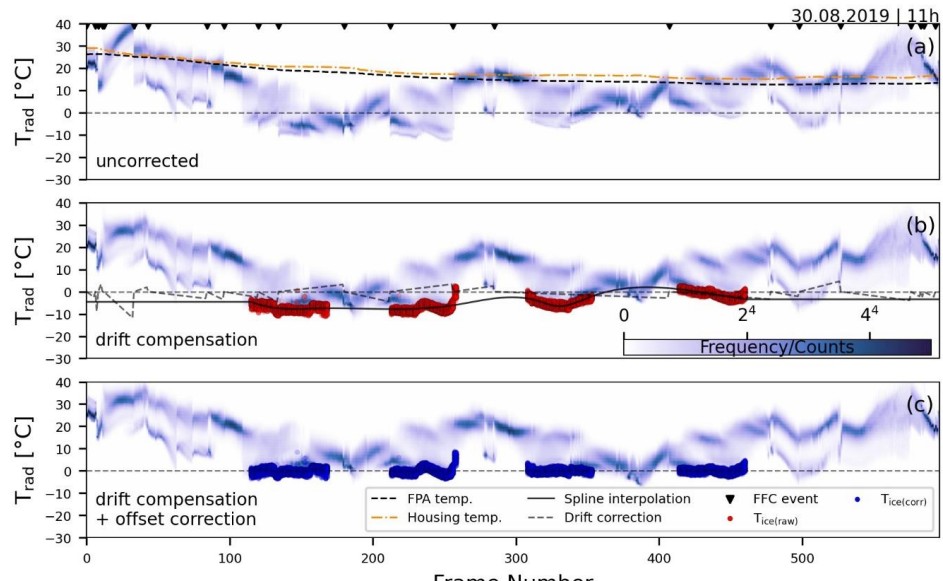

**Fig. 2. Thermal correction and calibration using bare ice temperatures on 2019-08-30 11:00 local time. Histogram timeseries, each vertical stripe shows the frequency of measured temperatures of a thermal infrared image. (a) Raw at-sensor (brightness) temperature with detected flat field correction events (black triangles) and thermal drift correction offset (dashed line). (b) Drift compensated temperatures with ice surface temperatures (red dots). Black line shows the spline interpolation of the measured ice surface temperatures with the edges set to a constant value. (c) Offset corrected temperatures, each frame based on the spline interpolation that the ice surface temperatures are at 0 °C.**

Despite successful detection of FFC events and applied drift compensation during postprocessing of the temperature data, we still observed bare-ice surfaces with considerable temperature deviations from 0 °C, the expected temperature for a melting ice surface. Furthermore, the remaining temperature bias appears to be not constant with time (Fig. 2b). Therefore, we applied a further calibration step that employs ice surface as a reference, asserting a 0 °C LST. The extraction of the ice surface was done by a color-based segmentation algorithm using k-means clustering (Pedregosa et al., 2011) and subsequently manually confirmed, similar to the approach of Aubry-Wake et al. (2015). We then interpolated the ice temperatures using splines and calculated an offset correction for each frame, in a manner that the LST of the ice will be 0 °C (Fig. 2c).

### 3.3 Orthomosaic generation (photogrammetry)

The generation of orthomosaic maps of both the optical and thermal image datasets (Fig. 1) was accomplished using the program Agisoft Metashape (Agisoft, 2020). The diurnal variation of the surface temperature and relatively low contrast of thermal images lead to spatiotemporal variations in the reconstruction of the 3D point clouds. Instead of additional point cloud alignment (Rusinkiewicz and Levoy, 2001), we orthorectified the thermal images using the same digital surface model (DSM) obtained from simultaneously recorded optical images. Therefore, we identified and marked all GCPs in both the optical and thermal images prior to the photogrammetric processing to improve the image alignment and improving the calculation of the camera calibration parameters (Cook, 2017). The generated DSM from the optical images was then used as the basis for the thermal image orthorectification.

### 3.1 Land surface temperature (LST)

The temperature measured by the sensor, the brightness temperature, is influenced by (1) the upward directed path radiance, (2) the radiation emitted by the surface towards the sensor and (3) the reflected portion of the incoming atmospheric longwave radiation. Due to the low flight elevation of 80 m above ground we neglect the path radiance. The reflected portion of incoming





atmospheric longwave radiation (3) was taken from the downward thermal flux of ERA5 Land hourly reanalysis data (Muñoz

Sabater, 2019) with respect to the time of flight. The large footprint of the reanalysis data (0.1° × 0.1°) compared to the small test site might introduce additional uncertainties. However, the influence on the LST is small, as the magnitude of the reflected radiation is also very small. The retrieval of the LST (2) is then a function of the emissivity of the surface material and the atmospheric transmissivity between the ground and the sensor. We assume the transmissivity to be negligible under the meteorological conditions and flight altitude (Kraaijenbrink et al., 2018; Malbéteau et al., 2018). Following Stefan-Boltzmann

law we calculated the LST using:

$$\text{LST} = \sqrt[4]{\frac{\sigma T_{rad}^4 - (1 - \varepsilon) \cdot LW \downarrow}{\sigma \varepsilon}}$$

Eq. 1,

where $\sigma$ is the Stefan-Boltzmann constant ($5.67 \times 10^8$ Wm$^{-2}$K$^{-4}$), $\varepsilon$ the emissivity of the surface type (debris=0.94, rough ice=0.97) (Rounce and McKinney, 2013; Aubry-Wake et al., 2015), and $LW\downarrow$ the incoming longwave radiation (Wm$^{-2}$). Some authors point out the relevance of the atmospheric transmissivity (Torres-Rua, 2017; Herreid, 2021), while others neglect it

due to low UAV flight elevations above the ground surface (D. G. Sullivan et al., 2007; Hill-Butler, 2014). We think our calibration procedure compensates for possible radiation attenuation by water vapor content in the atmospheric column. To assign emissivity values across the glacier surface, we distinguished between ice and debris using a supervised random forest classification with manually created training data (Breiman, 2001). Best classification results were found when the temperature differences between ice and debris were the largest, in the imagery obtained at 15 h.

### 3.2   Surface energy balance model

Radiative, convective, and conductive energy fluxes at the earth surface are described in the surface energy balance approach used here. For a layer of supraglacial debris, the rate of change of heat stored in the debris ($\Delta S$) must balance all incoming and outgoing energy fluxes (all fluxes have units of Wm$^{-2}$ and are positive when directed towards the debris layer):

$$\Delta S = SW + LW + LE + H + G$$

Eq. 2,

where $SW$ and $LW$ are the net shortwave and longwave radiation fluxes, $H$ and $LE$ are the sensible and latent heat fluxes, and

$G$ is the conductive ground heat flux. Parameters used in the surface energy balance are listed in Table 1.

**Table 1 Parameters used in the surface energy balance model**

| Model parameter | Symbol | Unit | Value |
|---|---|---|---|
| Debris albedo | $\alpha_d$ | - | 0.30 |
| Ice albedo | $\alpha_i$ | - | 0.64 |
| Debris emissivity | $\varepsilon_d$ | - | 0.94 |
| Ice emissivity | $\varepsilon_i$ | - | 0.97 |
| Effective thermal conductivity | $k$ | W m$^{-1}$ K$^{-1}$ | 0.96 |
| Surface roughness length | $z_0$ | m | 0.016 |
| Measurement height air temperature | $z_t$ | m | 2 |
| Measurement height wind speed | $z_u$ | m | 10 |
| Debris density | $\rho_d$ | kg m$^{-3}$ | 1496 |
| Debris specific heat capacity | $c_d$ | J kg$^{-1}$ K$^{-1}$ | 948 |
| Specific heat capacity of dry air | $c_a$ | J kg$^{-1}$ K$^{-1}$ | 1010 |
| Standard sea-level pressure | $P_0$ | Pa | 101325 |
| Air density at sea-level elevation | $\rho_{air}$ | kg m$^{-3}$ | 1.29 |

The net shortwave radiation is a function of the albedo and the amount of incoming solar radiation. We assumed a constant debris surface albedo of 0.3 (Rounce and McKinney, 2013; Schauwecker et al., 2015). Computation of the insolation at the

time of the UAV flights was done using a python implementation of the R package 'insol' (Corripio, 2003). The model





determines the solar geometry (Iqbal, 1983) and estimates the atmospheric transmissivities (Bird and Hulstrom, 1981) based on a digital elevation model, which in our case was generated from the optical UAV images. Atmospheric attenuation was calculated using the relative humidity and air temperature, from ERA5 Land hourly reanalysis data at each time of flight (Muñoz Sabater, 2019). We also accounted for cast shadows by the surrounding topography, based on a 0.5-m resolution

digital elevation model with larger footprint (Swisstopo, 2010).

Net longwave radiation results from the difference of incoming longwave radiation ($LW{\downarrow}$) and outgoing longwave radiation ($LW{\uparrow}$). $LW{\downarrow}$ is the same as in Eq. 1 and based on ERA5 Land data (see section 3.1). $LW{\uparrow}$ is a function of the LST and the surface emissivity (see section 3.1) and calculated following Stefan-Boltzmann's law $LW{\uparrow} = \varepsilon\sigma LST^4$. The latent heat flux (LE) is assumed to be 0, as the debris surfaces were dry during the UAV flights.

The sensible heat flux H was estimated using the bulk aerodynamic approach assuming a neutral atmosphere (Nicholson and Benn, 2006; Steiner et al., 2018; Nakawo and Young, 1982; Rounce and McKinney, 2013):

$$H = \rho_{air}\frac{P}{P_0}c_a C_{bt}u(T_{air} - LST) \qquad \text{Eq. 3,}$$

where $\rho_{air}$ is the air density at sea level pressure (kg m$^{-3}$), $P_0$ is atmospheric pressure at sea level (Pa) and $P$ atmospheric pressure at site elevation (Pa), calculated following Iqbal (1983), $c_a$ is the specific heat capacity of air (J$^{-1}$kg$^{-1}$K$^{-1}$) (Brock et al., 2010; Barry et al., 2021), $u$ the wind speed (ms$^{-1}$), $T_{air}$ the air temperature and $C_{bt}$ the bulk transfer coefficient given

as:

$$C_{bt} = \frac{k_*^2}{\ln(\frac{z_u}{z_0})\ln(\frac{z_t}{z_0})} \qquad \text{Eq. 4,}$$

where $k_*$ is the Kármán constant (0.41), $z_0$ the surface roughness length (Rounce and McKinney, 2013; Stewart et al., 2021) and $z_u$ and $z_t$ the measuring height (m) for wind speed and air temperature. The meteorological input data $u$ and $T_{air}$ were taken from ERA5 Land hourly reanalysis data (Muñoz Sabater, 2019).

The conductive heat transfer through the layer of debris and into the ice can be described by Fourier's law assuming a

homogeneous layer of debris:

$$G = -k\frac{\partial T}{\partial z} \approx -k\frac{LST - T_{di}}{d} \qquad \text{Eq. 5,}$$

where $\frac{\partial T}{\partial z}$ is the temperature gradient in the debris layer and k the effective thermal conductivity (W m$^{-1}$K$^{-1}$). We assume a linear temperature gradient in the debris layer and thus $\frac{\partial T}{\partial z}$ to be equal to the difference between the LST and the temperature of the debris-ice interface ($T_{di}$), which we assume to be at the melting point 0 °C. The assumption of a linear temperature gradient applies only approximately and for thin debris thicknesses (Conway and Rasmussen, 2000; Nicholson and Benn,

2006; Rounce and McKinney, 2013). The average diurnal temperature profile through a layer of debris can be considered linear but at sub-daily time intervals the profile varies in its degree of linearity (Reid and Brock, 2010). We will come back to this point in the discussion.

Solving the surface energy balance at sub-daily time intervals requires knowledge of the energy flux due to the change of heat stored in the layer of debris (ΔS) (Brock et al., 2010).

$$\Delta S = \rho_d c_d \frac{\partial \overline{T}_d}{\partial t}d \qquad \text{Eq. 6,}$$

where $\rho_d$ is the debris density (kg m$^{-3}$), $c_d$ is the specific heat capacity of debris (J kg$^{-1}$K$^{-1}$), $d$ the debris thickness (m) and $\frac{\partial \overline{T}_d}{\partial t}$ the average rate of mean debris temperature change (K s$^{-1}$) with $\overline{T_d}$ as the mean debris temperature, $(LST + T_{di})/2$, and $t$ the time. Our sub-daily multitemporal LST measurements allow us to estimate temporal changes in LST, but these are very sensitive to uncertainties in the LST measurements (see section 3.2). To avoid such issues, we rely on the diurnal temperature cycle and fitted a linearized harmonic sine function (Shumway and Stoffer, 2016) to the temperature data of each pixel. The




first derivative with respect to time of this function is the warming/cooling rate and can be used to calculate the change in the heat storage term.

### 3.3 Debris thickness estimation

As both the storage heat flux (ΔS) and the ground heat flux (G) in the surface energy balance model (SEBM) are a function of
the debris thickness, the surface energy balance model (section 3.5) can be described by a quadratic equation in form of:

$$d^2 \left( -p_d \, c_d \frac{\partial \overline{T}_d}{\partial t} \right) + d(SW + LW + H) - k(LST - T_{di}) = 0 \qquad \text{Eq. 7,}$$

Solving for debris thickness was done using the quadratic formula:

$$d = \frac{-b + \sqrt{b^2 - 4ac}}{2a} \qquad \text{Eq. 8,}$$

with $a = -\rho_d c_d \frac{\partial \overline{T}_d}{\partial t}$, $b = S + L + H$ and $c = -k(LST - T_{di})$. Note that the quadratic equation has mathematically two solutions whereas only one is physically plausible.

In addition to the SEBM approach, we also estimated debris thickness for each LST map using a rational curve (McCarthy,
2019; Boxall et al., 2021) of the form

$$d = \frac{LST}{c_1 + c_2 LST} \qquad \text{Eq. 9,}$$

where $c_1$ and $c_2$ are empirically derived coefficients by a least squares regression.

To evaluate the performance of the two approaches for predicting debris thickness, we used the RMSE between the predicted and the observed debris thickness at the sites surveyed in the field (section 3.1). Sites for which the SEBM approach did not yield a real and positive number were excluded from comparison. To evaluate the least squares regression of the rational curve
we divided the observed debris thickness data into a testing (n=45) and training (n=45) dataset. The testing dataset has been used to derive the model coefficients $c_1$ and $c_2$ while the testing dataset was used to compare the modelled debris thickness estimates with the filed observations.

### 4 Results

#### 4.1 Land surface temperature and its diurnal variation

The LST changes over the day in a cyclic manner (early morning cool – afternoon hot – evening cool) and consequently the relationship between LST and debris thickness changes accordingly (Fig. 3). Unlike satellite derived LST observations, our diurnal LST measurements allow us to show how this relationship changes throughout the day.

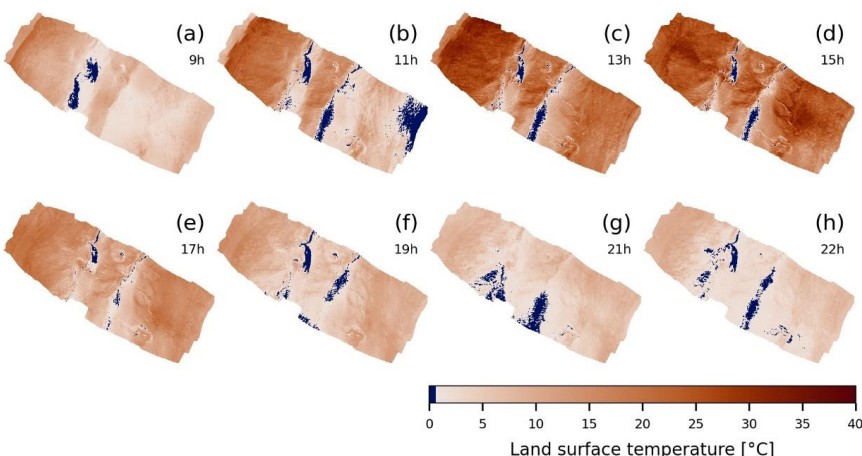

**Fig. 3. Spatiotemporal distribution of the land surface temperature (LST).** The panels (a-h) show the 8 individual flights describing
a large fraction of the diurnal land surface temperature variation. The maps have a spatial resolution of 15 cm and are colorized in
dark blue for LST < 0.5 °C. The south-east region of panel (b) (11 h) shows an unreasonable cold temperature due to a failed
calibration.

For generally cooler temperatures, LST appear to linearly increase with increasing debris thickness (Fig. 4a, b, f-h). In the

afternoon hours when the debris surface reaches its maximum diurnal temperature, the relationship between debris thickness

and LST shows its non-linear nature (Fig. 4c-e). Additionally, we observe the influence of the terrain aspect: east and south

facing slopes heat up earlier compared to west and north facing slopes (Fig. 4a, b) (Crameri et al., 2020).

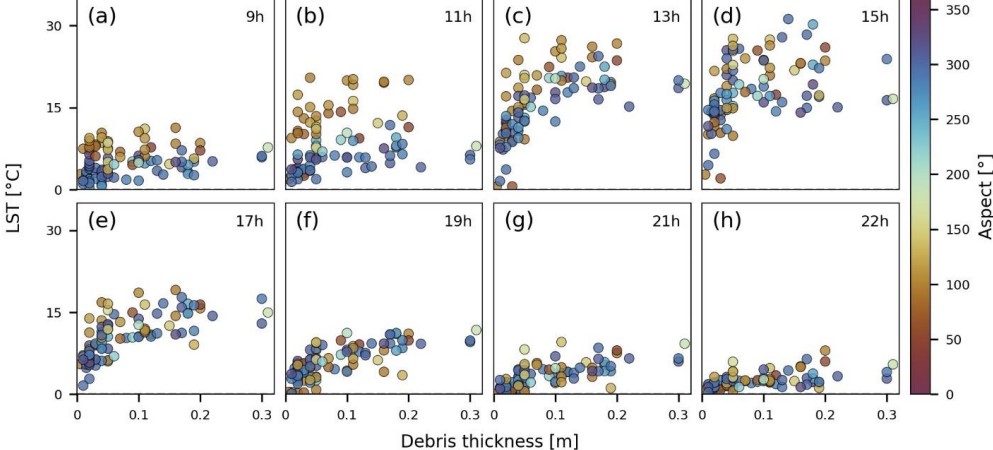

**Fig. 4. The temporal variation of the land surface temperature (LST) against in-situ measured debris thickness.** Panels (a-h) show
the arithmetic mean land surface temperature of a 2 m buffered region around the GPS coordinates of debris thickness
measurements colorized for terrain aspect. The LST of the warming phases (a-d) are stronger influenced by the aspect than the
cooling phases (e-h). The non-linear nature of the relationship of the LST and debris thickness is noticeably pronounced for higher
LST (c-e) while the relationship for low LST appears almost linear (f-h).

The effect of terrain aspect is not evident during the cooling phase in the afternoon and evening. The spatial and temporal

variability of the LST (Fig. 3) shows that at all flight times, surface temperatures are higher at the edges of the glacier (NW

and SE) and lower in the central part of the test area. This pattern corresponds to high debris thicknesses at the glacier margins

and thin debris thicknesses or no debris occurrence in the middle part. The mean LSTs of the debris cover follow the expected

pathway of a diurnal temperature cycle (Table 2).





**Table 2 Mean LST and standard deviation (1σ) of the debris and ice surface type.**

| Local flight time (h) | Mean ± 1σ (debris) °C | Mean ± 1σ (ice) °C |
|---|---|---|
| 09 h | 7.28 ± 4.60 | 1.71 ± 1.43 |
| 11 h | 10.68 ± 7.69 | 0.87 ± 1.87 |
| 13 h | 19.52 ± 7.13 | 1.43 ± 2.87 |
| 15 h | 21.44 ± 6.16 | 1.51 ± 2.32 |
| 17 h | 13.31 ± 4.46 | 2.26 ± 2.84 |
| 19 h | 8.10 ± 3.76 | 1.16 ± 1.56 |
| 21 h | 5.31 ± 2.80 | 1.31 ± 1.56 |
| 22 h | 4.59 ± 2.99 | 0.72 ± 0.77 |


While the general spatial and temporal pattern of LST seems to be reasonable, some areas of concern exist locally. First, the south-eastern region of the 11 h flight shows an unreasonable cold temperature patch, which most likely does not represent the actual LST at that time. Instead, we suspect that this artefact corresponds to an uncorrected bias of the thermal correction and calibration process. We will come back to this point in the discussion. Second, the 15 h flight shows a centrally located,

transverse oriented strip of higher LST, which seems to follow the flight path of the UAV. The directional temperature mismatch could be related to an oblique viewing angle of the sensor as nadir alignment was set up manually and the angle of observation might partly control the amount of radiation received by the sensor and thus the temperature measurement (Norman and Becker, 1995).

In the absence of any other means to assess the precision of the LST values, we suggest that the variability of the bare ice

surface temperatures, which ought to be at 0°C, might indicate the bias and precision of the LST. The LST values of ice surface temperatures vary by several degrees with a standard deviation of up to 2.87 °C (Table 2). Mean ice LSTs range between 0.72 and 2.26 °C throughout the day (Table 2). Field observations show that ice cliffs on TNG are often sprinkled with small rocks and/or a thin layer of dust, which might influence the ice LST towards warmer temperatures.

Based on the LST measurements, we estimated the diurnal variation of the depth-integrated mean debris temperature. The

pixel-wise fitted harmonic sine functions allow us to estimate the spatially distributed warming and cooling rates, as the first derivative with respect to time. The RMSE of the fit ($mean \pm 1\sigma$) is $1.51 \pm 0.54°C$. The spatial distribution of the RMSE (Fig. 5b) is relatively continuous but shows variability where the beforementioned local LST discrepancies occur. Fig. 5a shows that, depending on the aspect, at 13 h and 15 h the debris surface reaches its maximum LST and consequently the temperature change rate converges to zero.

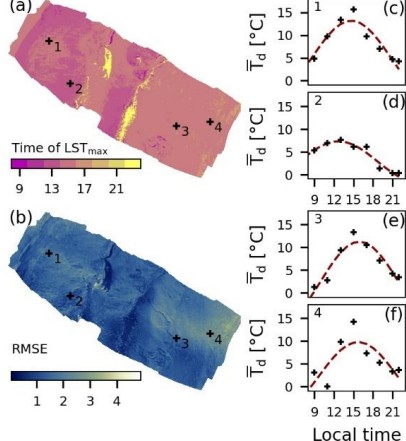


**Fig. 5. Sinusoidal regression of mean debris temperature ($\overline{T}_d$) for each pixel. (a) The panel shows the time at which $\overline{T}_d$ reaches its maximum diurnal temperature as a function of terrain aspect. (b) The RMSE of the regression for each pixel. The south eastern**



**region with larger errors is due to the anomalies in the LST at 11 h, see text for details. The spatial mean of the RMSE is 1.51 °C**
**and the standard deviation 0.54 °C. The panels (c-f) are example points of the sine function used to derive the warming/ cooling rate**
**in each pixel. Locations of the examples are indicated in panel (a) and (b).**

### 4.2 Surface energy balance modelling

To solve the surface energy balance (Eq. 2) we determined the LST-independent energy flux component $SW$ and the LST-
dependent components $LW$, $H$, $\Delta S$ and $G$ based on the UAV-derived LST maps shown in Fig. 3. In Fig. 6, we show the diurnal
variation of each component, evaluated at all locations where we obtained debris thickness measurements. To account for the
accuracy of the handheld GPS device, we used the mean variable values within a 2-m-radius buffered region around the GPS
coordinate.

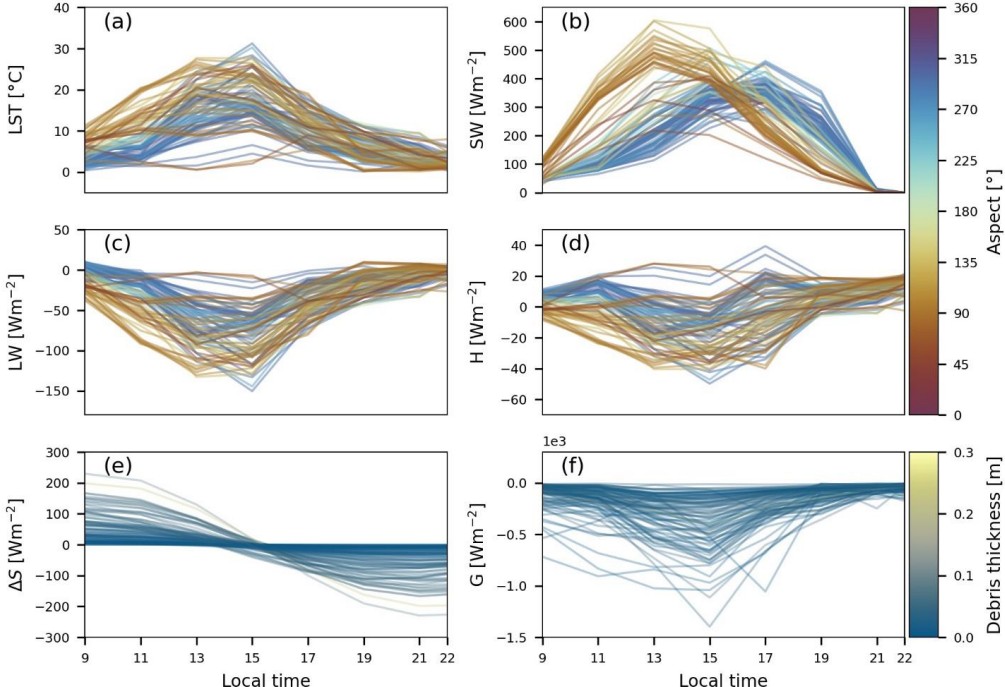

**Fig. 6. Energy fluxes at debris thickness measurement locations. Diurnal variations of (a) land surface temperature, LST, (b) net**
**shortwave radiation, $SW$, (c) net longwave radiation, $LW$, and (d) sensible heat flux, $H$, with lines colorized by terrain aspect. Diurnal**
**variations of (e) the change in heat storage, $\Delta S$, and (f) the ground heat flux, $G$, with lines colorized by debris thickness. Note that**
**only $SW$ (b) is independent of LST, whereas data in panels (c-f) are a function of LST.**

East and south facing slopes receive their maximum net shortwave radiation ($SW$) prior to west and north facing slopes (Fig.
6), which explains their earlier increase in LST (Fig. 6). By 15 h, all sites attained the daily maximum LST and cool down
from then on. Despite remaining differences in $SW$, no more aspect-related differences in LST can be observed. All the
remaining SEBM components are a function of the LST and thus also show an aspect dependency prior to 15 h. The net
longwave component ($LW$) expectedly mirrors the LST (Fig. 6c). The sensible heat flux ($H$), calculated using the bulk approach
(Eq. 3), attains only low flux values close to 0, which is likely related to the low wind velocities (<1 ms$^{-1}$) obtained from
reanalysis data (Table 3). The rate of change in heat storage within the debris ($\Delta S$) and the ground heat flux ($G$) are, besides
the LST, a function of the debris thickness (Eq. 6, Fig. 6e, f). Whereas $\Delta S$ attains the largest magnitudes in the morning and
evening hours and where the debris is thick, the opposite is true for $G$, which is largest at 15h and where the debris is thin.

**Table 3 ERA5 Land hourly reanalysis data on 30.08.2019 interpolated at Tsijiore-Nouve Glacier, Switzerland (46.01° N, 7.46° E)**

| Local flight time (h) | Incoming longwave radiation, $LW\downarrow$ (W m$^{-2}$) | Wind speed, $u$ (m s$^{-1}$) | Air temperature, $T_a$ (°C) |
|---|---|---|---|
| 09 h | 311.03 | 0.35 | 6.64 |
| 11 h | 304.38 | 0.45 | 10.29 |
| 13 h | 304.05 | 0.48 | 11.69 |
| 15 h | 307.31 | 0.85 | 12.09 |
| 17 h | 308.26 | 0.87 | 10.24 |
| 19 h | 307.46 | 0.41 | 8.98 |
| 21 h | 306.71 | 0.46 | 7.74 |
| 22 h | 306.19 | 0.60 | 7.20 |

### 4.3   Debris thickness estimates from SEBM

The SEBM-derived predictions of debris thickness (Fig. 7) show a general pattern that matches observations in the field and

the pattern of measured LST. Predicted debris thicknesses generally range between 0 and 30 cm. Given the chosen input

parameters, Eq. 8 cannot be solved for all pixels in the first half of the day (9 h to 15 h) (Fig. 7a-d).

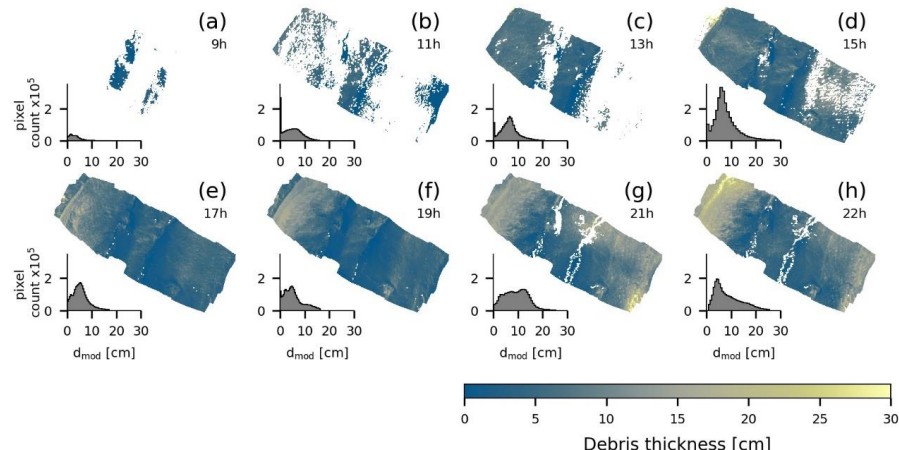

**Fig. 7. Estimated debris thicknesses for each flight time. White regions show regions where the surface energy balance model has no valid solution for debris thickness. The histograms show the distribution of the predicted debris thicknesses displayed in the maps.**

At these times the quantities of the surface energy balance components and the relatively low LST, lead to a negative term

under the root in Eq. 8, and thus to no valid solution. Predictions of thicker debris are primarily found in the afternoon and

evening hours (17 h to 22 h) and the pattern of thin debris predictions, primarily in the central part of the glacier, is relatively

consistent in time.

Comparing the predictions to field observations (Fig. 8) shows that the accuracy of the prediction remains comparable

throughout the day with a RMSE of 6 to 8 cm. For some of the flights the predictions correlate well with the observations,

even if they do not follow the 1:1 line. During the warming phase of the day, when aspect has a strong influence on LST (Fig.

4a-c), the associated debris predictions do not show an aspect-related pattern. In contrast, debris thickness predictions based

on the afternoon flights at 17 h and 19 h, seem to correlate with aspect whereas the LST data does not.





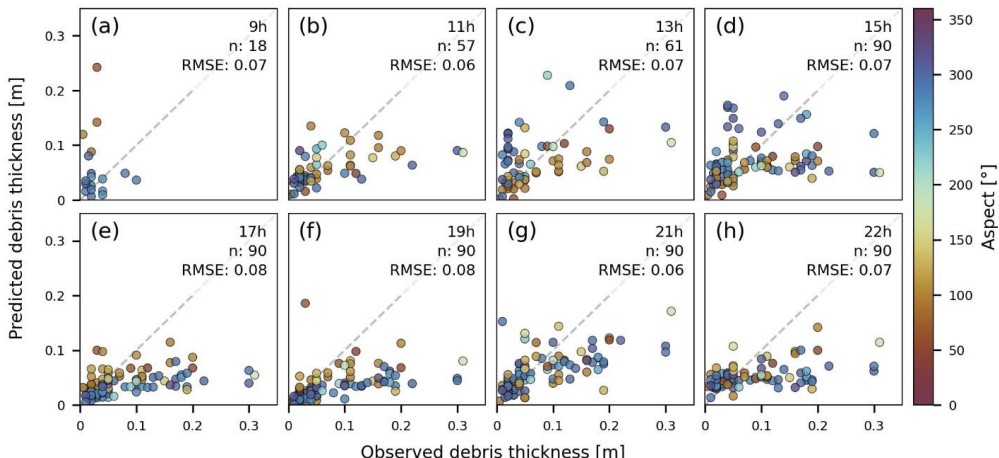

**Fig. 8. Comparison of modelled and observed debris thickness for each flight time (a-h) with RMSE values (m) for model evaluation. Sample points are colorized by terrain aspect and the gray dashed line shows the 1:1 line. RMSE for flight (a, b and c) is based on a reduced sample number.**

Predictions of thin debris cover are less sensitive to the time of the day, compared to thick debris. Figure 9a shows the variation of the predicted thicknesses with time along the profile introduced in Fig. 1b as the mean value $\pm 1\sigma$. Towards the glacier edges where the debris is greater than 10 cm, the spread in the standard deviation increases, compared to the central part, showing that the prediction of thick debris cover varies stronger in time than for thin debris cover.

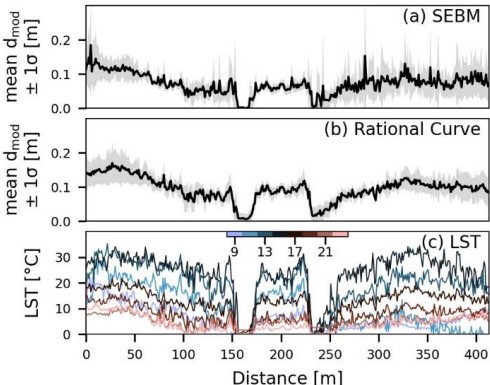

**Fig. 9. Diurnal mean ± 1σ debris thickness predictions along the profile line shown in Figure 1 (a) The predictions using the surface energy balance model (SEBM) show a larger spread of the standard deviation (gray) towards the edges and small towards the central part corresponding to regions of ticker and thinner debris cover. Panel (b) shows the results of the rational curve extrapolation approach. A smaller spread indicates greater consistency in the prediction over the day. (c) The diurnal variability of the land surface temperature along the profile line.**

### 4.4 Debris thickness estimates by extrapolating a rational curve

The debris thickness maps created by the extrapolation approach using a rational curve result in slightly thicker predictions than following the SEBM approach (Fig. 10). The general pattern of the spatial debris thickness distribution follows the field observations and the pattern of measured LST, similar to the results of the SEBM approach. Predicted debris thicknesses range between 0 cm and 30 cm and predictions of the first two flights at 9 h and 11 h, but do not exceed 10 cm (Fig. 10a, b).

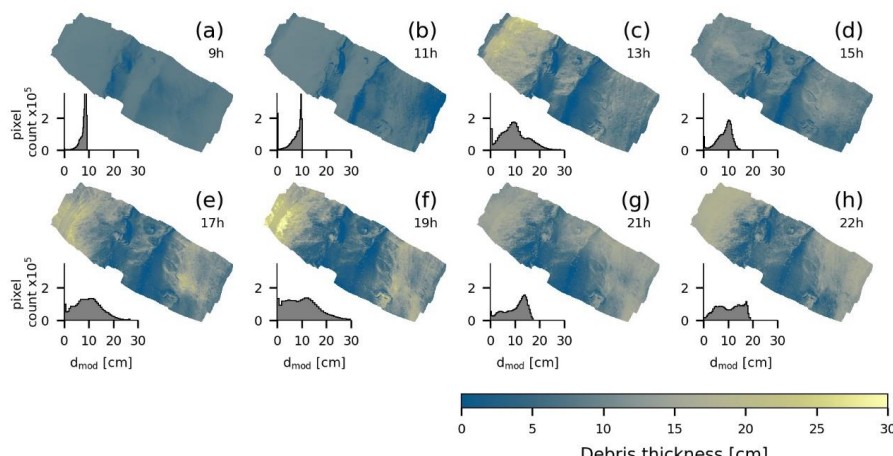

**Fig. 10. Estimated debris thickness using a rational function for each flight time. The histograms show the distribution of the predicted debris thicknesses displayed in the maps.**

We divided the dataset of n = 90 samples into a training (n = 45) and testing (n = 45) dataset. Fig. 11 shows training and testing data for each flight time including the coefficients $c_1$ and $c_2$, derived by least squares regression of Eq. 9 and the RMSE between the predictions and the observations of the testing data (Fig. 11). Similar to the SEBM, the RMSE ranges between 6 cm and 8 cm, but debris thicknesses > 10 cm are better represented in the extrapolation approach and thus follow more closely the 1:1 line. At 9 h, 11 h and 15 h the RMSE is highest with 8 cm and the shape of the curve already shows that the model does not represent the data well. The aspect dependency of the LST at these times (Fig. 4, a,b) was not considered as an additional parameter for the regression and thus, results in a curve that does not represent the shape of the data well (Fig. 11a,b). The afternoon flights between 17 h and 22 h have the lowest RMSE with 6-7 cm.

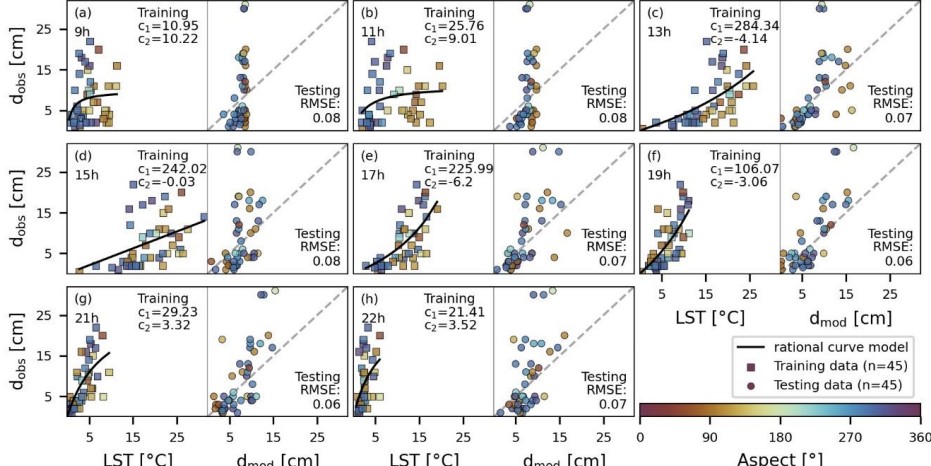

**Fig. 11. Least squares regression of a rational function of 50 % of the debris thickness measurements (n=45) with the arithmetic mean land surface temperature of a 2 m buffered region around the GPS coordinates of debris thickness measurement locations (training data) (a-h). The adjacent panels show the comparison of modelled and observed debris thickness of the other 50 % (n=45) with RMSE values to evaluate the prediction (testing data). Sample points are colorized by terrain aspect.**

The diurnal stability in predicting debris thickness (Fig. 9. Diurnal mean ± 1σ debris thickness predictions along the profile line shown in Figureb) shows that thin debris cover, as found in the central part of the profile line, remains stable throughout



the day and is thus comparable to results of the SEBM approach. For thicker debris the spread of the standard deviation is higher, showing that predicting thick debris cover depends more on the time of the day that thin debris. Even though the range

of the RMSE throughout the day remains comparable to the SEBM results, for some flights (e.g., 19 h) the average prediction accuracy improves about 2 cm.

## 5    Discussion

### 5.1    Predicted versus observed debris thickness

Debris thickness predictions with the SEBM approach yielded mixed results. The fact that the modelled debris thickness does

not vary in unreasonable ways across the glacier surface, but in a systematic pattern shows that mapping high resolution debris thickness with UAVs is viable. During most of the flights, we observe a general positive correlation between the modelled and observed debris thickness (Fig. 8b-h). The overall relationship between higher surface temperature and thicker debris that is evident in the input data (Fig. 4) can be reproduced. However, given the chosen parameters, we are unable to obtain a non-biased match between the observed and modelled debris thickness. The SEBM approach mostly underestimates debris

thickness at all flight times. For thick debris cover this underestimation is more pronounced than for thin debris (<10 cm).

Over the course of the day the RMSEs between observed and predicted debris thicknesses range from 6 to 8 cm. For many pixels in the flights at 9 h, 11 h, 13 h and for some pixels at 15 h (Fig. 7Fig. 7 a-d) the quadratic equation (Eq. 7) has no real solution, and the SEB cannot be solved for debris thickness. The reason for this is a negative term in the square root of Eq. 8, which occurs if $b^2 < 4ac$. Recall that $b$ accounts for the radiative and sensible heat fluxes ($SW+LW+H$), whereas $a$ and $c$ are

the ground heat flux and the storage term, respectively. As long as LST > 0 °C, $c$ is always negative. The inequality condition above can thus only occur if also $a$ is negative, which is only possible when the debris is heating up, in our case until about 15 h (Fig. 5a, Fig. 6e). That explains, why many pixels in the morning flights have no debris thickness solution. Furthermore, at 9 h, the term $b^2$ is rather small, mostly because of low $SW$ values. However, at 11 h and 13 h southeast-exposed pixels receive higher $SW$ (Fig. 6b), which causes $b$ to increase, making the inequality condition less likely.

As a result of spatially incomplete debris thickness maps, the number of sample points to evaluate the quality of the prediction is reduced (see low 'n' in Fig. 8a-c) and should be kept in mind when comparing the RMSE with respect to the time of the day. Previous studies did not face this issue, in part because $\Delta S$ was incorporated as a fraction of the ground heat flux $G$, which is always negative (Foster et al., 2012; Schauwecker et al., 2015). Comparison of $\Delta S$ and $G$ for the sites where we measured debris thickness shows that such an assumption appears to be invalid for most times of the day (Fig. 5). By estimating $\Delta S$ using

the warming/cooling rate from multitemporal LST measurements, we can better account for this energy balance component, but these estimates are also prone to uncertainties in LST. In general, however, the magnitude and distribution of $SW$, $LW$, $\Delta S$ and $G$ for most of the sample locations compare well to values determined by Brock et al. (2010) with an automatic weather station (AWS) at the Miage Glacier at comparable latitude, time of the year, and elevation (500 m difference).

Debris thickness predictions below ~10 cm seem to correlate reasonably well with field observations, whereas predictions of

thicker debris cover are generally too low (Fig. 8). This may indicate that uncertainties in parameters that are unlikely to vary spatially or as a function of debris thickness are not particularly relevant. To further test this hypothesis, we performed sensitivity tests of SEBM-derived debris thickness estimates to variations in the input parameters air temperature, wind speed, thermal conductivity, albedo, and surface roughness length (Fig. 12). Variations of the parameters air temperature, albedo, and surface roughness length across value ranges commonly found in the literature (Brock et al., 2000; Foster et al., 2012;

Schauwecker et al., 2015; Shaw et al., 2016; Miles et al., 2017) result in generally small variations of the mean debris thickness (averaged across the entire studied surface). In consequence, the impact on the RMSE when evaluated against our field observations of debris thickness are also small. Only the flights at 9 h, 11 h, and 13 h show more significant variations in the





RMSE, but these correspond to simultaneously low coverage of valid predictions and thus only small numbers of sample points to estimate the RMSE (Fig. 9. Diurnal mean ± 1σ debris thickness predictions along the profile line shown in Figure).

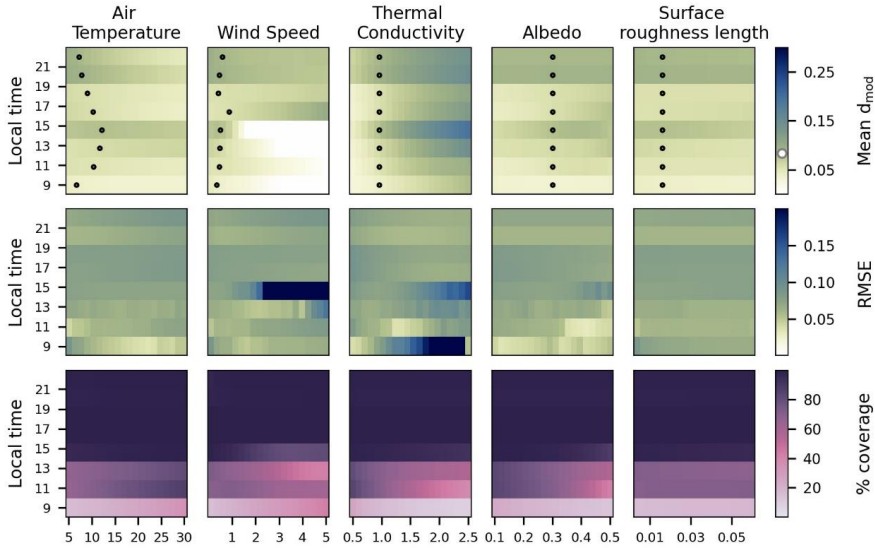


**Fig. 12 Sensitivity of debris thickness prediction using a surface energy balance model (SEBM) to the parameters air temperature, wind speed, effective thermal conductivity, albedo and surface roughness length. For each flight time, each parameter was varied across a range of values and debris thickness maps were created. Each column shows colorized (1) the modelled mean debris thickness averaged over all pixels in the test area, (2) the RMSE between the observed and predicted debris thicknesses and (3) the**
**coverage of valid predictions as the surface energy balance model cannot always be solved. The white dot in modelled mean debris thickness shows the mean debris thickness observed in the field.**

The same is essentially true for wind speed, which is a notoriously difficult parameter to constrain in any surface energy balance model (Schauwecker et al., 2015; Stewart et al., 2021) and is a strong control of the sensible heat flux (Eq. 3). ERA5-derived wind speed during the time of our experiment is relatively low at <1 m s$^{-1}$, whereas wind speeds of ~2-4 m s$^{-1}$ are not
uncommon in the vicinity of glaciers (e.g., Oerlemans and Greuell, 1986; Brock et al., 2010; Steiner et al., 2018). During a different visit to the TNG in 2021, we operated a small AWS for a full day and obtained wind speeds of 2.5-4 m s$^{-1}$, which were higher than ERA5-derived wind speeds of 0.5-2 m s$^{-1}$ for the same day. Although we don't know what the actual wind speed was during our experiment in 2019, increasing the wind speed and solving for debris thickness has a minor effect on flights from 17 h onwards, whereas for earlier flights, the coverage quickly drops to low values. This is related to the fact that
larger negative $H$ values reduce $b$ (i.e., the sum of the radiative and sensible heat fluxes, $SW+LW+H$), thereby increasing the likelihood to obtain a negative term in the square root of Eq. 8. Similar effects also account for changes in coverage for the other parameters. We thus emphasize that changes in RMSE during flights until about 15 h that are associated with changes in coverage do not necessarily indicate better model performance.

The only tested parameter that has a more pronounced effect on the mean debris thickness and RMSE without changing the
coverage is the thermal conductivity ($k$) through its influence on the ground heat flux, $G$. Higher $k$ values result in greater energy losses to the ice and a higher debris thickness for the same LST. A similar effect has been achieved by Rounce and McKinney (2014) through introducing a factor to account for the non-linearity of the temperature profile in the debris cover. It should also be noted that the effective thermal conductivity $k$ is likely to vary spatially, as thick debris cover can hold more moisture, which thus leads to higher values of $k$ (Steiner, 2021). Additionally, a thin debris cover composed of smaller grain
sizes may have different pore space than a layer of thick debris cover consisting of larger grain sizes. The bulk debris-void space and thus the effective conductivity could vary with debris thickness, too. Because the effective thermal conductivity of





a debris layer and its spatial variability is a rather complex quantity that is not easily measured, this parameter could be used as a free parameter to tune the debris thickness map against field observations.

Debris thickness predictions using least squares regression of a rational curve yield similar and in parts more accurate results compared to the SEBM approach. The pattern of spatially distributed debris thickness estimates (Fig. 10) follows the expected spatial pattern of the LST for each time of the day. The range of predicted debris thicknesses correspond to the field observations with values similar to the SEBM approach, between 0 cm and 30 cm. As the relationship of LST and debris thickness varies throughout the day, the suitability of the rational curve regression varies too. For instance, at 9 h and 11 h, LST depends strongly on the terrain aspect and thus the results are biased towards aspect (Fig. 11a,b). Nevertheless, at 13 h

the RMSE between observations and predictions (Fig. 11c, testing data) is still 7 cm. This suggests that during the times when the debris is heating up, the regression of a rational curve would benefit from taking terrain aspect into account, i.e., by fitting a parametric surface to the data. In addition, the strongly non-linear relationship between LST and debris thickness at 9 h and 11 h limits predicted debris thicknesses to <10 cm. The predictions at these flight times show unrealistic uniform values in the same regions at which the SEBM approach cannot be solved. This supports the SEBM approach and indicates that the LST at

this time is too low to relate it to debris thickness. When the debris is cooling down in the evening (Fig. 11e-h), aspect has a minor effect, and the curve appears to satisfy the available data. When LST is low (morning and evening), the relationship between LST and debris thickness seems to be almost linear, and a simple linear regression is expected to result in comparable accuracy. This agrees with findings of Boxall et al. (2021), based on satellite derived LST.

### 5.2   Opportunities and limits of UAV-derived LST for debris thickness mapping

High resolution studies can improve our understanding of processes that move and distribute debris on glacier surfaces. Multitemporal UAV-derived debris thickness maps would thus allow to quantify how debris is mobilised across the glaciers surface over short times scales. The detailed representation of surface features, such as ice cliffs, large boulders (Fig. 13), or surface ponds make UAV-derived LST measurements a valuable tool for debris cover research.

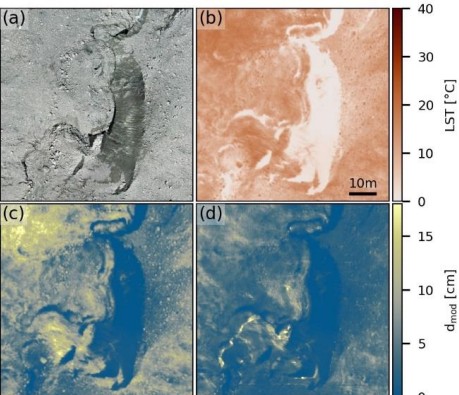

**Fig. 13. High resolution subsection of the unpiloted aerial vehicle imagery (UAV) with debris cover, large boulders and ice cliff. The panels show (a) the RGB image, (b) the land surface temperature at 15 h, (c) debris thickness prediction using the rational curve approach and (d) debris thickness prediction by solving the surface energy balance model.**

So far, the conversion of LST to debris thickness in high resolution was only studied using ground-based oblique viewing angles (Herreid, 2021) using empirical equations. However, challenges with (1) area covered by the field of view, (2) the

variable path radiance and (3) the viewing angle than controls the amount of radiation received by the sensor. UAV offers opportunities, but also faces challenges that we discuss here. These challenges stem from the specifics of image acquisition, post-processing, and the conversion of the brightness temperature to LST.



All high-resolution studies, including ours, have so far used uncooled microbolometers, a sensor type that requires thermal equilibrium between the sensor device and the environment for accurate measurements (Budzier and Gerlach, 2015). As these

conditions are difficult to achieve and maintain in high mountain settings, the obtained thermal infrared images require calibration and correction. The ambient temperature difference between the ground and the flight elevation requires the sensor device to thermally adjust after take-off, which thus introduces a measurement bias that varies with time (Fig. 2a). While this effect is primarily relevant for UAV applications, the maintenance of stable environmental conditions (e.g., changing wind speeds) cannot be guaranteed even for ground-based measurements and temporal variance of the measurement bias should be

considered. While the sensor device cools down after take-off due to flight altitude, direct incident shortwave radiation may cause the device to heat up (Dugdale et al., 2019). The change in the measurement bias with time, the thermal drift, is partially balanced by the internal in-flight calibration of uncooled microbolometers (Mesas-Carrascosa et al., 2018), leading to recurring systematic jumps in the measured temperature (Fig. 2) that can be compensated for in a postprocessing step (see section 3.2). Long flight times, slow flight speeds and no direct shortwave radiation would thus minimize the effect of thermal drift but

would likely not substitute for additional calibration.

The thermal correction during post-processing in our case included (1) recovering the occurrence of flat field correction (FFC) events that were "lost" by the reduced sampling rate, (2) identifying and correcting the thermal drift, and (3) correcting the residual measurement bias using bare ice surfaces. During all flight times, thermal drift corrected for by FFC events was rather large and resulted in changes by up to ~8 K over 50 frames (Fig. 2a). With a frame rate of 1/s, this means a thermal drift of up

to 0.16 K s$^{-1}$. Assuming that the thermal drift is indeed linear with time, the in-flight FFC or, as in our case, post-processing identification of FFC is relatively straight forward, due to the step change in LST across an FFC event. Figure 2a also shows the internal housing temperature and the temperature of the focal plane array, as recorded by the thermal sensor. The rapid decline in the beginning shows the thermal adjustment due to the vertical temperature gradient between the ground and flight elevation. While UAVs with larger battery capacity might offset this effect to some extent, our setup was limited in that point.

To convert the brightness temperature to LST, we accounted for the reflected portion of the incoming longwave radiation and surface type emissivity but neglected the path radiance between the sensor and the ground. As the elevation of the UAV above ground does not change significantly throughout the flight, the potential measurement bias of longwave radiation emitted by atmospheric water vapor content is minimized and assumed to be constant. Our 'bulk' calibration approach using spline interpolation of measured ice surface temperatures compensates for the systematic temporal variability of the measurement

bias (Fig. 2b) introduced by (1) thermal adjustment after take-off, (2) fluctuations of atmospheric conditions by wind or incident direct shortwave radiation or (3) longwave radiation emitted from atmospheric water vapor content or the surrounding terrain (Aubry-Wake et al., 2015; Aragon et al., 2020; Herreid, 2021).

Because of the need to calibrate all thermal images, the requirement of spatially well distributed reference temperatures is the main drawback of the proposed method. In our case, bare ice surfaces were present in the central part but not at the glacier's

sides. Two image regions are found to be severely erroneous, an anomalously low temperature patch on the eastern edge at 11 h and a warm temperature stripe in the centre that seem to follow the flight path of the UAV at 15 h (Fig. 3b,d). We think the cold region at 11 h could be due to a failed drift compensation, as the spline interpolation assumes a constant correction value before the first and after the last occurrence of bare ice in the thermal images. The warm temperature strip could be related to an oblique viewing angle of the sensor during that flight, as the sensor alignment was done manually (Sobrino and Cuenca,

1999; Byerlay et al., 2020; FLIR, 2020).

So far, we discussed the challenges and needs to derive glacier surface LST's. Provided the measurements obtained in our experiment, we also observed differences in the resulting debris thickness that we derived from the SEBM and the rational curve approaches. The SEBM approach requires meteorological input data, assumptions on debris properties (in space and time) and substantial simplifications of SEB components. Because conservation of energy represents a balance among all

energy fluxes, it follows that any simplification in one component will have a quantitative effect on the others (Price, 1985).





However, when comparing the diurnal variation of the energy flux components with measured quantities in a comparable setting regarding location, time, and debris thickness (Brock et al., 2010), we find good agreement in magnitude and distribution for net shortwave, net longwave and change in heat storage and the ground heat flux (Fig. 6). The possibility to estimate sub-daily surface energy balance components improves our understanding of $\Delta S$. Repeated LST measurements might additionally increase understanding of the spatial variability of debris properties (e.g., thermal conductivity, debris density or specific heat capacity) by quantifying the thermal inertia.


The accuracy of predicting debris thickness using a SEBM and empirically using a rational curve yielded comparable results with a RMSEs of 6-8 cm depending on the time of the day. Both methods yield a terrain aspect bias. The SEBM approach compensates for the terrain effect to some degree as the amount of incident shortwave radiation is a function of aspect, too.


The terrain bias in the early flights using the rational curve approach is more pronounced as it is only based on the LST. Steep moraines of debris or hummocky shaped debris-covered surfaces are likely to introduce bias via mixed-pixel effects, when predicting debris thickness using coarse spatial resolution LST from remote sensing data, especially in the case of empirically derived debris thicknesses. For example, the time of overpass of the Landsat satellite is typically between 10 h to 11 h locally, a time when debris cover is still heating up. Therefore, the effect of aspect on satellite derived LST debris


thickness estimates should be studied in more detail.

## 6    Conclusions

In our experiment we mapped supraglacial debris cover using high-resolution UAV-derived LST measurements at various times of the day and using two common approaches to create debris thickness maps: a surface energy balance model approach and a simple extrapolation approach using a rational curve that relies on field measurements. We conclude that:


1) Measuring the LST from an UAV using an uncooled microbolometer requires temperature calibration that varies with time. Here we determine an offset correction value for each thermal infrared frame by interpolating splines of spatially well distributed bare ice surfaces, assuming the ice to be at melting point 0 °C. This bulk correction compensates for several sources of uncertainties, but requires the presence of bare ice surfaces.


2) Quantifying the surface energy balance components based on the UAV-derived LST measurements led to debris thicknesses predictions with a RMSE of 6-8 cm, depending on the time of the day. Debris thicknesses were underestimated at all flight times. Measuring the diurnal variability of LST allowed to extend the commonly used surface energy balance approach by quantifying the rate of change of heat storage.

3) The non-linearity of the relationship between LST and debris thickness increases with LST. Choosing the best empirical function for predicting debris thickness thus depends on the time of the day. Morning conditions yield a strong terrain aspect bias, which is better accounted for in the SEBM approach. When the LST reaches its diurnal maximum, here at 13 h or 15 h, the non-linearity is most evident. Towards the evening the relationship between debris thickness and LST appears almost linear and aspect plays a minor role.


4) Practical considerations for quantifying supraglacial debris cover using UAV-derived LST comprise LST calibration, choosing the model based on the time of the day and debris thickness measurements for evaluation. In our case, the ultra-light weight UAV set-up was suitable for remote high mountain field work but had a significant drawback due to the limited battery capacity, resulting in short flight times of 10 to 15 minutes and small spatial coverage. Consequently, the thermal adjustment of the device led to strong thermal drift and thus to many in-flight calibration events that had to be considered. Maximizing the flight time by using a larger UAV could offset this effect to some degree. The measurement bias varies with time and spatially well distributed reference temperatures should be used for calibration.





## 7 Code availability

The source code and input files of the surface energy balance model, and the debris thickness data are available from Gök et al. (2022), Repository:

https://dataservices.gfz-potsdam.de/panmetaworks/review/f56042df84c0684384b1a1ce8171d3e488b234d46cb4fd6b3fb9b2abf545a112/

**Author contributions**

The study was designed by DTG and DS. DTG developed the hardware set-up and conducted UAV flights. LSA conducted
debris thickness measurements. DTG performed the analysis with support from DS. DTG wrote the initial version of the manuscript. DS and LSA commented on the initial manuscript and helped improving this version.

## 8 Funding

This research received funding from the European Research Council under the European Union's Horizon 2020 research and innovation programme under grant agreement 759639.

## 9 Acknowledgements

We are grateful to K. Wetterauer and D. Dennis for field support, to T. Walter for advice with the drone equipment, and to Marcel Ludwig for help in assembling the hardware setup.

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
