# Peer review of "High-resolution debris cover mapping using UAV-derived thermal imagery: limits and opportunities"

_The Cryosphere, 2022_

## Referee Comment (RC2)

Review of: High-resolution debris cover mapping using UAV-derived thermal imagery: limits and opportunities, Gök et al.

**General comments:**

Gök et al. present a comprehensive study investigating the derivation of debris thickness from thermal imagery at different times throughout a day using two approaches: solving of a SEBM using reanalysis data, and a least squares regression method that utilises in-situ debris thickness measurements. This study advances knowledge by investigating the diurnal changes in debris thickness estimations and addresses the impact of thermal drift due to the use of uncooled microbolometers by applying a manually calibrated drift compensation. My line specific minor comments to improve clarity in the manuscript are below and any major issues to be addressed are in bold.

**Specific comments:**

**ABSTRACT**

L10: Specify what LST you use (i.e. LST over debris layer)

L15: Can you specify which method is the most appropriate.

L18: Please state recommendation for which time of day best represents the most accurate debris thickness estimations.

**INTRODUCTION**

L22: I would disagree with the statement that debris thickness is generally rather thin, debris thickness is highly variable! – please rephrase and/or add a supporting reference.

L26: Change heavily to extensively or similar word.

L37: Provide references for these processes – e.g. Kirkbride and Deline, 2013; Hartmeyer et al. 2022a,b.

L38: Provide reference for 'debris thickness varies with time' and 'recent studies document changes…'

L43: Reword sentence for clarity.

L52: Add Gibson et al. 2017 as reference for empirical estimation of debris thickness using in situ debris thickness measurements.

L61: Westoby et al. 2020 use UAV optical imagery to estimate debris thickness and changes, so perhaps rephrase to emphasize that debris thickness estimation using UAV thermal imagery has remained elusive.

**STUDY AREA**

L70: Units for coordinates.

L76: Detail the specific dimensions of the study area here and replace 'numerous' with the specific number of flights.

**MATERIALS AND METHODS**

L90: Change pre-define to pre-defined.

L92: Can you state the resolution of your imagery here too? I.e. … has a resolution of 640x512 pixels, equating to a thermal image resolution of XXm x XXm over the study site.

L103: Add a statement that reflects the inherent bias of making in-situ manual hole measurements (i.e. the presence of a large boulder/ thick debris makes it less likely that it will be chosen it as a spot to dig).

L108: Can you provide a bit more detail about what unbalanced thermal conditions are? Spell it out for the reader.

L119: Why is there a reduced framerate?

L130: Define FPA acronym used in figure in caption.

L137: Change 'asserting' to 'assuming' – this is an assumption that you have made for the final calibration, the ice surface may be 0°C but ice surface temperatures can be colder than 0°C and this should be acknowledged. It is also apparent that after your correction, some ice surfaces are now pushing 10°C (frame number ~250 (c)) where the spline interpolation was not super effective, this should be explained and acknowledged too.

L145: State number of thermal and optical images used in this sentence.

L155: State size of test sight so comparison with the footprint of the reanalysis data is possible.

L166: Do you mean 'by *accounting* for the water vapor content…'?

L168: Why/how are they the best classification results?

L169: Add statement to end of sentence along the lines of 'thus data from this time stamp were used to classify the thermal imagery'.

L172: I'm not sure I understand how $\Delta S$ is a rate of change if the right hand side of Eq. 2 is fluxes in W m$^{-2}$?

L198: Can you comment on the accuracy of wind speed data taken from reanalysis products and the impact this will have on your subsequent debris thickness estimates (see Schauwecker et al 2015; Stewart et al 2021)?

L203: State what your definition of 'thin' debris thickness is.

**L229: Can you please justify why the debris thickness was estimated by solving a quadratic rather than previously documented methods such as that in Rounce and McKinney, 2014? The exclusion of sites for which there is not a real or positive solution to the quadratic equation means that a large proportion of your data is excluded from further analysis – which poses quite a large problem when your study area is already quite small.**

L230: Change testing to training.

**RESULTS**

L236: I think it is the estimation of debris thickness that changes rather than the relationship between LST and debris thickness, no? i.e. when hotter LSTs are observed, thicker debris will be estimated?

**L240: A key problem for me with this figure is the lack of consistency between the areas of ice and debris in each subplot – theoretically, these areas should remain consistent throughout the day, the time scale of this study is not large enough to observe actual**

**change in the cliff geometries. This then throws into question the accuracy of the data in each time stamp if cliffs are not consistently detected. Can you provide an explanation / justification for this?**

L243: Can you show a linear regression line and an R2 value on each subplot to support this statement?

L270: Are these LST ranges using raw LST or offset corrected LST?

L295: In caption (or next to the color bar in the figure) state what aspect degrees refer to (i.e. N S E W). Also, is the debris thickness manually measured debris thickness? Make this clear.

L310: To support this statement, include a histogram of manually measured debris thicknesses for comparison with the modelled debris thicknesses in Fig. 7.

**L312: I am concerned about how much data is not valid in Fig7a-d, and I'm not convinced that 'no valid solution' is a sufficient explanation for the lack of data. A surface energy balance model should not be unsolvable. To compare data from different time stamps, the data needs to be (and should be) spatially consistent.**

L318: Quantify 'pattern of thin debris predictions'.

L343: Sentence beginning 'Predicted debris thicknesses…' does not make sense.

**DISCUSSION**

L374: underestimates compared to what?

L406: Figure units! X axis and also mean debris thickness in top right.

L417-421: My takeaway from this is that 1) wind speed is not modelled well with ERA-5 data, 2) if wind speed is increased to 'realistic' values then the amount of 'valid' debris thickness pixels decreases significantly? Can you discuss what implications this has in terms of the methodology? I.e. would you recommend that SEBM are not used in conjunction with thermal data to estimate debris thickness?

L434: Can you quantify 'in parts more accurate'?

L451-452: Westoby et al 2020 do this with optical imagery and a geodetic based debris thickness estimation.

L455: It would be good to see a debris thickness difference map (i.e. rational curve – sebm debris thickness) to quantify the differences between the two methods as panels c and d visually look very different. Are the differences between the two methods significant on a pixel by pixel basis?

L512-520: I would like to see a recommendation of 1) which method is the better predictor of debris thickness, and 2) at which time of day the method appears to be the most accurate. This seems to be missing from the discussion and the paper in general.

**Technical corrections:**

Ensure LST and not LST's throughout the paper.

Rounce and McKinney, 2013 – should this be Rounce and McKinney, 2014, see reference below?

L150: Check section numbers here and throughout (i.e. where they're referenced, such as in L187).

L362: Figureb?

L364: that thin debris should be than thin debris.

**REFERENCES**

Gibson, M.J., Glasser, N.F., Quincey, D.J., Mayer, C., Rowan, A.V. and Irvine-Fynn, T.D., 2017. Temporal variations in supraglacial debris distribution on Baltoro Glacier, Karakoram between 2001 and 2012. *Geomorphology*, *295*, pp.572-585.

Hartmeyer, I., Delleske, R., Keuschnig, M., Krautblatter, M., Lang, A., Schrott, L. and Otto, J.C., 2020. Current glacier recession causes significant rockfall increase: the immediate paraglacial response of deglaciating cirque walls. *Earth Surface Dynamics*, *8*(3), pp.729-751.

Hartmeyer, I., Keuschnig, M., Delleske, R., Krautblatter, M., Lang, A., Schrott, L., Prasicek, G. and Otto, J.C., 2020. A 6-year lidar survey reveals enhanced rockwall retreat and modified rockfall magnitudes/frequencies in deglaciating cirques. *Earth Surface Dynamics*, *8*(3), pp.753-768.

Kirkbride, M.P. and Deline, P., 2013. The formation of supraglacial debris covers by primary dispersal from transverse englacial debris bands. *Earth Surface Processes and Landforms*, *38*(15), pp.1779-1792.

Rounce, D.R. and McKinney, D.C., 2014. Debris thickness of glaciers in the Everest area (Nepal Himalaya) derived from satellite imagery using a nonlinear energy balance model. *The Cryosphere*, *8*(4), pp.1317-1329.

Schauwecker, S., Rohrer, M., Huggel, C., Kulkarni, A., Ramanathan, A.L., Salzmann, N., Stoffel, M. and Brock, B., 2015. Remotely sensed debris thickness mapping of Bara Shigri glacier, Indian Himalaya. *Journal of Glaciology*, *61*(228), pp.675-688.

Stewart, R.L., Westoby, M., Pellicciotti, F., Rowan, A., Swift, D., Brock, B. and Woodward, J., 2021. Using climate reanalysis data in conjunction with multi-temporal satellite thermal imagery to derive supraglacial debris thickness changes from energy-balance modelling. *Journal of Glaciology*, *67*(262), pp.366-384.

Westoby, M.J., Rounce, D.R., Shaw, T.E., Fyffe, C.L., Moore, P.L., Stewart, R.L. and Brock, B.W., 2020. Geomorphological evolution of a debris-covered glacier surface. *Earth Surface Processes and Landforms*, *45*(14), pp.3431-3448.

---

## Author Comment (AC1)

**Responses to Reviewer 1, Sam Herreid**

TC-2022-113: "*High-resolution debris cover mapping using UAV-derived thermal imagery: limits and opportunities*" by Gök, et al.

*Thank you kindly for taking the time to carefully read our manuscript and for the constructive comments which helped us to improve it. In the following, we will address all comments point by point.*

"High-resolution debris cover mapping using UAV-derived thermal imagery: limits and opportunities" by Gök et al. uses thermal data collected during eight repeat UAV flights throughout one day of a portion of Tsijiore-Nouve Glacier in Switzerland to evaluate estimates of debris cover thickness. The authors use two established methods to estimate debris cover, but move research on this subject forward by their thorough consideration of thermal data drift and offset, the continued advancement of using thermal cameras on UAVs, and the repetition of flights over the course of one diurnal cycle. The authors do a nice job evaluating the debris thickness estimates against field measurements, however, I think a key deficiency in the results are debris thickness difference maps showing the variability of a quantity, which, if estimated perfectly, should be a grid of zeros. Even integrating to show the change (i.e. error) in total debris volume for this study domain might, for example, help future erosion rate studies looking to use debris thickness changes as a way to quantify bedrock erosion to understand the magnitude of error in these methods. Below are mostly minor, but some more major in line comments.

**ABSTRACT**

L10: be explicit: e.g. supraglacial debris surface temperature or LST over a debris cover

> *Changed to: "… we present land surface temperatures (LST) of supraglacial debris cover …"*

L11: can you say at x cm resolution rather than high?

> *Changed to: "… measured from an unpiloted aerial vehicle (UAV) at high (15cm) spatial resolution."*

L16: Can you make here in the abstract a statement on which method you found to be most efficient successful/efficient if the RMSE is essentially the same for both? This could be formulated as a recommendation for future studies.

> *Added sentence: "Although the rational curve approach requires in-situ field measurements, the approach is less sensitive to uncertainties LST measurements*

compared to the SEBM approach. However, the requirement of debris thickness measurements can be an inhibiting factor that supports the SEBM."

L16: This isn't exactly true that diurnal variability in LST controls the relationship between LST and debris thickness. Maybe you mean to say the success of a rational function to express the relationship varies predictably with time of day? Maybe also conditionally with clouds cover and precipitation?

Good point. We changed the sentence to: "Because LST varies throughout the day. the success of a rational function to express the relationship between LST and debris thickness also varies predictably with the time of day."

L20 This last sentence is unclear, do you mean independent measurements of LST or a better camera on your UAV? And also a little bit of a downer. Glaciologists can't practically afford a cooled microbolometer and they certainly aren't going on UAVs anytime soon. I would either make a more achievable suggestion or try a more positive closing sentence.

Uncooled microbolometers are generally able to perform temperature measurements with sufficient accuracy. However, insulating the device from influential factors in high mountain regions (e.g. sensor cools down after UAV take-off, wind, or direct shortwave radiation) is challenging. But we think there is room for improvement to perform more accurate measurements using this sensor type. We changed the sentence to: "… which are challenging to achieve with uncooled sensors in high mountain landscapes."

**INTRODUCTION**

L23: Add citation(s) to first sentence, Scherler et al, 2018: Herreid and Pellicciotti, 2020

Done.

L23: Add citation supporting "debris cover is generally rather thin, usually less than a meter"

We added the reference to Rounce et al., 2021

L24: I would change "profound" to something like "exponentially compounding influence …where melt rates are accelerated…" or at least "it can have a profound"

Changed to: … "it can have a profound" …

L26: by "heavily" do you mean spatially or in thickness?

Both. To be clearer we changed the sentence to: "Consequently, glaciers with widespread and thick debris cover can persist longer at lower…"

L29: I think it is established that the advancing you cite here is occurring more because of elevation and precipitation rates, not as a function of debris cover.

> *We agree and change the sentence to: "…with some glaciers being stationary and some retreating."*

L33: inappropriate location for citations

> *We agree and moved the citation to the end of the sentence.*

L39: Cite the recent studies

> *Added: Kaushik et al. 2022*

L39: There are practical reasons why there are more studies on extent change rather than thickness change, it's an easier problem with a better control on errors, it's intermediate studies like this one that may slowly help add the z component to change analyses.

> *Yes, we agree with this view. Yet, we don't feel this needs to be added to the text.*

L43: The list followed by "vary rapidly" reads poorly, maybe you mean the abundance of these features can increase or decrease rapidly or the abundance can vary dramatically between different glaciers or different parts of one glacier?

> *Good point. Changed to: "In particular, the abundance of supraglacial streams, ponds and ice cliffs can increase or decrease rapidly across the glacier surface (Anderson et al., 2021)."*

L45-46: what does "distribution" constitute beyond knowledge of "extent and thickness"? Also, a continuous model of debris thickness is more likely, not comprehensive observations.

> *Indeed, "distribution" is redundant and was removed. Although we agree that model-derived debris thickness maps are more likely than observation-based, we think the development of such models will always require comprehensive observations for evaluation.*

L49: Incorrect citation to Ostrem 1959, this paper established the relation between debris thickness and melt rate.

> *Citation removed.*

L49: In (3) explain where the debris thickness comes from, a residual from the methods you describe.

> *We changed the sentence to: "…the estimation of sub-debris melt by DEM differencing and converting melt rate to debris thickness based on the Østrem-curve, …"*

L52: I think you might mean "fitting … to" rather than "extrapolating … using"

*Changed accordingly*

L57: What do you mean by recent technological advances? Uncooled microbolometers have been around for quite a while. One limitation may be that it's a single sensor imaging a single wavelength window rather than a split window approach some thermal satellite sensors use to solve for and remove atmospheric attenuation.

*Good point, we changed the sentence to: "Most LST-based approaches to estimate debris-cover thickness have focussed on satellite imagery, whereas studies employing near-ground image acquisition in high resolution are less frequent."*

L60: What are the opportunities and limits? What is the research question? And what is the desired resolution, acquisition frequency and look angle? Some ideas of these are needed to motivate this statement.

*We removed this sentence at this point and focus on the discussion on opportunities and limits.*

L60: I think it's more "approaches of acquiring a thermal image" rather than "applications"

*Changed accordingly*

L61-62: the physics and methodologies seem the same to me whether the thermal imagery is acquired via oblique imagery or an airborne sensor. I would say what remains elusive is mapping high-resolution thermal for entire glaciers or several glaciers. A UAV is a good candidate for achieving this, although still a step beyond the scope of this paper.

*We agree and change the sentence to: "Debris thickness was recently mapped using oblique LST (Herreid, 2021), but the quantification of debris thickness from UAV thermal imagery has remained elusive."*

L64 and throughout LST not "LST's"

*Changed accordingly*

**STUDY AREA**

L70: coordinate units

*Changed accordingly*

**MATERIALS AND METHODS**

L102: I understand this, but maybe say point measurements are ambiguous or difficult to make in an unbiased/representative way.

> Changed to: "The debris cover close to lateral moraines consists of very large boulders (>0.5 m) that rendered measurements impractical and thus introduced a bias on the point measurements."

L103: how does this boundary look in your debris thickness estimates? Does this mean there were areas of >30 cm debris, just not sampled?

> Unfortunately, the mentioned large boulders made measurements impossible. Based on Figure 10 f, we observed the thickest DC along the north-western margin of the ice. However, as written, it is not entirely clear where the ice stops and the moraine begins. In terms of sampling: yes, especially where large blocks and boulders impeded digging into the debris, the debris thickness is likely larger than 30 cm and we did not sample this.

Section 3.2: I think the authors did a really nice job here illustrating a problem that future glaciologists might encounter and giving a solution to it. Is the spline, rather than a median value shift, capturing spatial variability in drift?

> Yes, the measurement bias at the beginning of a flight can be different that at the end of a flight. If there are well-distributed reference temperatures (in our study ice cliffs), then the spline seems to do a good job. However, at the beginning and the end of the flight, where no ice cliffs were in the field of view, we added a constant value like the median value approach.

I don't think you said yet in the paper how long a flight path took to fly, do we assume Fig. 2 is roughly one snapshot in time? Was this approach applied to each of the flights?

> We added a sentence in section 3.1: "Each flight took between 12 and 15 minutes and captured around 600 thermal images."

Was drift and offset roughly similar for each flight?

> Yes, therefore we decided to show only one example. The evening flights showed less variations of ice surface temperatures. We also added this information to the text of section 3.2: "The drift and offset were similar for most flights, but the evening flights showed less variation in the ice temperatures."

It seems like the bare ice temperature in (c) is still not 0C? . I trust that the air temperature during the survey was well above freezing and indicative of bare ice being at the pressure melting point, but it might be worth noting that there's nothing initially alarming about ice being less than 0C. Consider changing the section title to something like thermal image drift and offset correction.

> Yes, the variation of ice surface temperatures within one frame is quite large. This in-frame variation is not captured in the calibration procedure. The title changed to: "Thermal drift and offset correction"

Section 3.3: The thermal images were stitched and then orthorectified or orthorectified and then stitched? From Fig. 1 I'm guessing the number of frames is around 600 for each flight, can you give the number here. You say that you identified the GCPs in the thermal images but there are only 6 GCPs so most thermal images have no reference, this makes me guess that you stitched the thermal images first. How did you manage overlap? Did you take a mean or trim one image? It's clear from the section above you applied some corrections for drift and offset, but still I'm guessing there are residuals and the temperature may be slightly different for some of the overlap just due to temperature change during the gap time between image acquisition paths.

> We understand that more information on the image acquisition and mosaic generation is welcome and changed the text to:
>
> "Each flight yielded around 600 thermal infrared frames (Fig. 2), of which around 400 have been used to generate orthomosaic maps and 200 were omitted as they recorded the take-off and landing of the UAV. The diurnal variation of the surface temperature and relatively low contrast of thermal images led to spatiotemporal variations in the reconstruction of the 3D point clouds. Instead of additional point cloud alignment (Rusinkiewicz and Levoy, 2001), we orthorectified the thermal images using the same digital surface model (DSM) obtained from simultaneously recorded optical images. Therefore, we identified and marked all GCPs in both the optical and thermal images prior to the photogrammetric processing to improve the image alignment and improving the calculation of the camera calibration parameters (Cook, 2017). As the footprint of the images is relatively large with respect to our area of interest, the 6 GCPs were visible in almost all thermal images. The generated DSM from the optical images was then used as the basis for the thermal image orthorectification. The overlapping parts were reduced by a weighted average during the orthomosaic generation. Agisoft Metashape software offers several options on how to handle overlap areas and we found the default setting to produce the most reasonable results. "

L150: watch section numbering

> Changed accordingly

L166: you are making the assumption that atmospheric attenuation is linear, but still I think I agree that this is a negligible term in this study.

> Yes, we agree.

L171: Convective heat transfer was more 'deemed not present' than included

> Good point! We changed the start of the sentence to: "Thermal energy fluxes…"

L173, Eq. 2: All fluxes have units of Wm^-2 so delta-S must as well, how is this a rate of change of heat, where is the time derivative coming from?

It's the rate of change in heat storage within the debris (e.g., Brock et al, 2010). To balance surface energy fluxes at sub-daily time intervals, the debris layer must be treated as a volume with variable heat storage. It requires knowledge of the average rate of mean debris temperature change, which we obtain by fitting a sine function to the diurnal LST variation in each pixel of the map and forming the first derivative for each time of flight (described in L220-L225 and examples shown in Fig. 5). We explain the delta-S in more detail later in this section with Eq. 6 and think this is sufficient. Please, let us know if there is still something unclear.

L175: What exactly is the ground in this context?

The base of the debris layer. We expanded the sentence by: "… from the debris into the underlying ice." The expression ground heat flux is often used for the conductive heat flux through the layer of debris. But to be more precise we changed ground heat flux to conductive heat flux throughout the manuscript.

L185: with a larger

Done.

L219: stored heat flux

Done.

L221: where did d in the second term come from?

Following Brock et al. (2010) we describe the rate of change of heat storage in Equation 6, where the debris thickness is included. Consequently, the SEBM now contains G(d) and ΔS(d) which then results in a quadratic equation when solving for debris thickness. Previous energy balance approaches to solve for debris thickness did not face this issue as ΔS was either not considered or roughly approximated.

L229: Fig. 7 suggests this null area of unphysical SEBM results is nontrivial, this seems like a very notable detraction, a lot of future analyses that will use debris thickness maps will need to be continuous. The lack of a meaningful solution in areas with debris also raises concern about the area where there is a solution.

Yes, we agree with the reviewer that these unphysical results are relevant, especially in the morning flights. We also agree that these uncertainties may raise concern for areas with existing solutions. We devoted much of chapter 5.1 to discussing and explaining the cause of this issue. In short, such issues arise specifically when the debris is heating up during the morning to midday when uncertainties in LST or reanalysis-derived quantities can cause no solution to the equation. In addition, we address the impact of several parameters on the model and the unphysical results (sensitivity test Figure 12). As the second reviewer had similar concerns, we expanded the discussion section in 5.1 on these issues in the revised version. At this point in the text, we refer the reader to the discussion section but refrain from

getting into too much detail before showing results. Added sentence: "We discuss the causes for these unphysical solutions in detail in section 5.1."

L230: I think you mean training

Changed accordingly

**RESULTS**

L241: How many scenes make up the corner of (b) that is incorrect? The geometry of the erroneously cold area seems odd. Do you think the >0.5C data is reliable nearby? I'm curious why the bare ice / ice cliff geometry isn't consistent between images? That would be a data quality red flag if stable thermal features aren't captured in a repeatable fashion.

Around 20 scenes. The issue likely lies in the calibration being less reliable towards the beginning and end of the flights when no ice cliffs are available for calibration.

Thanks for pointing out the inconsistency in ice cliffs – that's an important point! The inconsistency in the bare ice geometry is due to the residual uncertainties of the ice surface temperature. Figure 2 shows that the ice surfaces can vary by several degrees (~6 K). This means that our threshold of 0.5 K in figure 3 does not capture all ice surfaces. We added to the caption of figure 3: "Due to the residual uncertainties of the LST, ice surface geometries appear inconsistent in time."

L247: Is 0/360 degrees north?

Yes. Changed figure caption to: "… colorized for terrain aspect with 0/360 ° facing north."

L280: Fig. 5. The plot of time of LST_max is really clever! But how is it a function of terrain aspect? Wouldn't it be a combination of aspect and the thermal inertia of the debris? Isn't bar{T_d} just LST/2 per definition in this paper (LST + 0)/2 )(L211)? Does this quantity really have more meaning than LST alone?

Thanks for the credits! We agree that the LST is a function of both aspect and thermal inertia. The influence of aspect is nicely shown in the figure and examples. The influence of thermal inertia is more difficult. As thermal inertia is a material property, it may change across the scene. However, our impression during fieldwork was that the rock type making up the debris layer does not vary across the surface. In any case, what we show in the figure is only the effect of aspect, which does not mean that thermal inertia plays no role. To make it clear that we only refer to the influence shown in the figure, we changed the sentence in the caption to: "The panel shows the time at which $\bar{T}_d$ reaches its maximum diurnal temperature and thereby emphasizes the effect of terrain aspect."

It is true that $\bar{T}_d$ is directly connected to the LST. As we don't know the mean debris temperature, we approximate $\bar{T}_d$ using the LST. We added an explanation to the figure caption, but we prefer to keep the figure as it is, because it connects

immediately with Eq. 6, which describes the computation of delta-S. Caption text changed to: "Sinusoidal regression of mean debris temperature ($\overline{T}_d$), estimated from LST (**Fehler! Verweisquelle konnte nicht gefunden werden.**) for each pixel..."

L289: Somewhere can you add a histogram of debris thickness please? Fig. 6(e,f) suggest many of the measurements were 5cm or less. L101 said the mean was 9cm but maybe also state the median? Or perhaps a different drawing order in the fig or different data visualization if there are too many lines to show the whole dataset. I'm just concerned that the methods used here are better suited (meaning more stable for method evaluation) to slightly thicker debris.

That's a great suggestion! We added a histogram of the field measurements to figure 1. We agree the median information would be helpful too. The sentence changed to: "The debris cover on the TNG is generally thin: measured thicknesses are below 30 cm, with a mean of 9 cm, a median of 5 cm and a standard deviation of 10 cm."

L290: methods not results

True, we moved the sentence to section 3.6

L305: I don't get this, are you saying G is actually large for thin debris or just a very subtle max over a time series of low values? G might be sensitive to inaccurate LST over thin debris cover because the ice should keep the numerator of G low or near zero, but a thin debris thickness denominator might make G sensitive to LST error. Debris thickness measurements can also be quite sensitive over thin debris covers, where getting the local mean thickness wrong by 1 cm could be a high percentage error, while a 3 cm error over a 50 cm debris cover will propagate less error though these sorts of calculations.

These are good points, thanks! In fact, the absolute values of G can be large. The values on the y-axis are scaled by 1e3, which was unfortunately hidden above the figure. In the revised version, we made sure that the values are shown on the axis. Yes, we absolutely agree with the reviewer. G is sensitive to inaccurate LST, which is also one of the reasons why we obtain unphysical solutions. We expanded our discussion in section 5.1 by the following:

"The most likely reason for no physical solution to Eq. 7 are inaccurate values of LST and reanalysis-derived variables. Mostly during the morning, even small deviations from true values are sufficient to find no physically meaningful debris thickness solution. For thin debris (< 2cm), G is very sensitive to uncertainties in LST and leads to large negative numbers. This is a major drawback of the SEB approach, and it highlights the sensitivity of the approach to uncertainties in the input data. We note that these uncertainties prevail, even if a solution is found. However, compared to the ground observations of debris thickness the model predictions show a positive correlation."

L316: I don't find this sufficient. Reality doesn't return an undefined solution, I think if the model fails and you understand why, you should write a piecewise function that either returns continuous results or at least sets constraints on where the method is applicable.

> We understand that discontinuous maps are not satisfying. However, we think it is an interesting behavior of the model that highlights some difficulties that might be hidden otherwise. By extending the common SEBM approaches by the component of ΔS, we introduce more complexity as now both G and ΔS are a function of debris thickness. SEB approaches as applied by e.g. Rounce and McKinney, 2014 or Foster et al., 2012 would result in continuous maps, which does not necessarily mean that these results are closer to reality. We are aware of the high uncertainties of our LST measurements and the reanalysis data, however, the shape of the diurnal temperature cycle (Fig 5 c, d, e, f) still shows the expected diurnal variation which is encouraging. We decided to take advantage of the sub-daily time intervals and show the discrepancies in the model results, which we also discuss in detail (section 5.1).

L313: Fig, 7: If the debris thickness estimates were continuous over the same domain the histograms would be informative, but with data voids they aren't very comparative. I would like to see difference maps, with respect to perhaps one reference time that agreed particularly well with measurements. The main challenge in this problem is taking variable input datasets and returning a constant value. Based on the images alone in Fig. 7 it looks like a general trend is preserved but the pixel-to-pixel debris thickness will be shown to vary quite a bit unrealistically over hours and I also see what look like artifacts in the data. Likely the same artifacts that are discussed in the text but I would be hesitant to keep them for use in the later analysis of this paper. It's possible the quality of thermal sensor that can fit in a DJI UAV isn't quite high enough to return clean data.

> We agree on the relevance to evaluate the debris thickness maps with respect to time. We already addressed this topic in Fig 9, by showing the mean +- 1σ of a profile line. We tested the reviewer's suggestion and created difference maps, with respect to the 21h flight time. However, we have difficulties picking a reference time for the difference maps and also don't want to overload the manuscript with two additional large figures (SEBM and rational curve). Therefore, we suggest expanding Figure 9 with a map of the standard deviation of the predicted debris thickness values and a map that shows the number of valid solutions of the SEBM to evaluate the consistency in time.

> Option 1:

[Figure]

Debris thickness difference [cm]

Option 2:

[Figure]

L317-318: I would like to read more quantitative results than "relatively consistent in time"

Changed to: "Predictions of thicker (>10 cm) debris are primarily found in the afternoon and evening hours (17 h to 22 h) and the pattern of thin debris (<10 cm) predictions, primarily in the central part of the glacier, is relatively consistent in time."

Line 320: Line 101 states the mean debris thickness is 9 cm so a RMSE of 6 to 8 cm means nearly 100% error for the average case.

Yes, we agree, and starting this research we wished for a better match between observed and modelled debris thicknesses.

L320-325: Fig, 8. I think this is just about how well debris thickness can be predicted from mostly or exclusively thermal data, and I think one should be careful from trying to see what we all wish it would show. "Correlate well with observations even if they do not follow the 1:1 line" sounds like seeing what you want to see, a successful predictive model really should be clustered around the 1:1 line.

These are valuable points! We agree with the reviewer and rephrased the sentence to: "For most of the flights we find a positive correlation between the predictions and observations, even if they do not follow the 1:1 line."

L328: I'm not sure it's so clear about the thin debris, if you change the plot limits to 5 cm I think it might look like random scatter. The distance away from 1:1 might be less, but with respect to the magnitude of the debris thickness there may not be clear evidence of predictive capability. This tight cluster of thin debris cover near the origin likely also weights the RMSE favorably. If you consider only debris greater than 5 cm thick, for example, the RMSE will likely increase.

These are valid points. In the sentence this comment referred to, however, we address how sensitive predictions of thin debris are to the time of day: "Predictions of thin debris cover are less sensitive to the time of the day, compared to thick debris." Although the percentage change may indeed be still large, the absolute change in predicted thin debris cover is smaller than that for thick debris. To make this clearer, we changed the sentence to: "Absolute values of predicted thin debris cover are less sensitive to the time of the day, compared to thick debris."

L335: Are you computing the standard deviation from 7 or 8 measurements? Is that enough?

Well, we see the point of the reviewer. However, we show the standard deviation simply to indicate the variability, and for this purpose, we think it serves the case.

L347: How did you divide the dataset?

We used the python scikit learn function: sklearn.model_selection.train_test_split()
https://scikit-learn.org/stable/modules/generated/sklearn.model_selection.train_test_split.html

Citation added to the text.

**DISCUSSION**

L370: comma after pattern

Done.

L371: I think you need to show how well the model results are consistent in time as well as approach the testing dataset to make a statement about method viability.

> Good point. We toned down the sentence to: "The fact that the modelled debris thickness does not vary in unreasonable ways across the glacier surface, but in a systematic pattern shows that mapping high-resolution debris thickness with UAVs has some potential."

L405: Fig. 12: x-axis units

> Done.

L445: this could be the piecewise condition mentioned above

> See answer to comment L316

L450-452: I think this might be true with a more advanced methodology, but the results from this study show, what look like, large changes in debris thickness over a day when the quantity should be found as static. Mapping high-contrast features like ice cliffs from high resolution thermal data has already been shown to be useful, but studies relating changes in debris thickness to erosion rates or surface properties will need quite high confidence debris maps, since the changes are likely to be on the order of cm per decade.

> Yes, we agree with the reviewer that the results from this study do not allow us to study changes in debris thickness as the uncertainties in LST are too high to resolve small changes. However, we think that this study gives an orientation where the difficulties arise and discuss in detail the limits and opportunities of UAV thermal image derived debris thickness estimations.

L460: the high emissivity of rocks limit reflected radiation making the viewing angle less of a factor, but still can be accounted for. The angularity of rock clasts make even a nadir look-angle non-normal to most surfaces.

> We agree.

L462: I would add battery limitations on range as well.

> Done.

L475: What about a more expensive camera as well? I assume higher end FLIR uncooled microbolometers have a higher accuracy even in less ideal conditions.

> Probably yes, but we think the more relevant point would be a better in-flight calibration (e.g. with a mobile blackbody) that would lead to higher accuracies.

---

## Author Comment (AC2)

**Responses to Reviewer 2**

TC-2022-113: "*High-resolution debris cover mapping using UAV-derived thermal imagery: limits and opportunities*" by Gök, et al.

We would like to thank reviewer #2 for the comments and suggestions that helped us to prepare an improved revised version of our manuscript. Please find our responses in blue below your comments.

General comments:

Gök et al. present a comprehensive study investigating the derivation of debris thickness from thermal imagery at different times throughout a day using two approaches: solving of a SEBM using reanalysis data, and a least squares regression method that utilises in-situ debris thickness measurements. This study advances knowledge by investigating the diurnal changes in debris thickness estimations and addresses the impact of thermal drift due to the use of uncooled microbolometers by applying for a manually calibrated drift compensation. My line specific minor comments to improve clarity in the manuscript are below and any major issues to be addressed are in bold.

Specific comments:

**ABSTRACT**

L10: Specify what LST you use (i.e. LST over debris layer)

> Good point, changed to: "In this study, we present land surface temperatures (LST) of supraglacial debris cover and its diurnal variability …"

L15: Can you specify which method is the most appropriate.

> Added sentence: "Although the rational curve approach requires in-situ field measurements, the approach is less sensitive to uncertainties in LST measurements compared to the SEBM approach."

L18: Please state recommendation for which time of day best represents the most accurate debris thickness estimations.

> We added: "We find the rational curve approach with LST measured in the evening hours to yield the best result."

**INTRODUCTION**

L22: I would disagree with the statement that debris thickness is generally rather thin, debris thickness is highly variable! – please rephrase and/or add a supporting reference.

We added a reference to Rounce et al., 2021

L26: Change heavily to extensively or similar word.

Changed to: "Consequently, glaciers with widespread and thick debris cover can persist longer at lower elevations than debris-free glaciers …"

L37: Provide references for these processes – e.g. Kirkbride and Deline, 2013; Hartmeyer et al. 2022a,b.

Done.

L38: Provide reference for 'debris thickness varies with time' and 'recent studies document changes…'

We changed the first sentence to "As all these processes vary with time, supraglacial debris cover ought to change in time, too. " We did not add references to the sentence "Indeed, recent studies document changes in debris cover thickness in various mountain ranges on Earth.", because these are provided in the following sentence: "Most studies, however, focus on changes in the extent of debris cover (Shukla et al., 2009; Bhambri et al., 2011; Glasser et al., 2016; Tielidze et al., 2020, Kaushik et al., 2022), whereas studies documenting changes in thickness are relatively rare (Stewart et al., 2021; Gibson et al., 2017)."

L43: Reword sentence for clarity.

Very true, we changed the sentence to: "In particular, the abundance of supraglacial streams, ponds and ice cliffs can increase or decrease rapidly across the glacier surface (Anderson et al., 2021)."

L52: Add Gibson et al. 2017 as reference for empirical estimation of debris thickness using in situ debris thickness measurements.

Done.

L61: Westoby et al. 2020 use UAV optical imagery to estimate debris thickness and changes, so perhaps rephrase to emphasize that debris thickness estimation using UAV thermal imagery has remained elusive.

Good point. Changed to: "Debris thickness was recently mapped using oblique LST (Herreid, 2021), but the quantification of debris thickness from UAV thermal imagery has remained elusive."

**STUDY AREA**

L70: Units for coordinates.

Changed to: "46.01 °N, 7.46 °E"

L76: Detail the specific dimensions of the study area here and replace 'numerous' with the specific number of flights.

> Changed to: A relatively small study area of 60000 m² allowed for 8 UAV flights covering the entire study area throughout the day.

**MATERIALS AND METHODS**

L90: Change pre-define to pre-defined.

> Done.

L92: Can you state the resolution of your imagery here too? I.e. … has a resolution of 640x512 pixels, equating to a thermal image resolution of XXm x XXm over the study site.

> Changed to: "… has a resolution of 640x512 pixels, equating roughly to a thermal image resolution of 0.17 m x 0.16 m and measures longwave …"

> The UAV flight path was terrain adjusted using a SRTM DEM to maintain the same distance to the ground. However, as the SRTM DEM has a coarser resolution, the elevation cannot be maintained exactly. This results in variation of the raw thermal image resolution. During the photogrammetry processing, differences in spatial resolution are canceled out.

L103: Add a statement that reflects the inherent bias of making in-situ manual hole measurements (i.e. the presence of a large boulder/ thick debris makes it less likely that it will be chosen it as a spot to dig).

> Changed to: "The debris cover close to lateral moraines consists of very large boulders (>0.5 m) that rendered measurements impractical and thus introduced a bias on the point measurements."

L108: Can you provide a bit more detail about what unbalanced thermal conditions are? Spell it out for the reader.

> Changed to: "Unbalanced thermal conditions between the inner parts of the sensor and the ambient temperature (e.g. the sensor cools down after UAV take-off, changing wind conditions, or heats up by direct incident shortwave radiation) introduce a temperature bias."

L119: Why is there a reduced framerate?

> We reduced the frame rate to facilitate data management as the default setting records thermal images in a framerate of 9Hz. We reduced the framerate to 1 image per second, which we considered to be enough. We were not aware of the consequences during the post processing using the Teax Thermal Capture 2.0 and the accompanying software ThermoViewer 3.0.7. The tau2 in-flight calibration is documented in the image metadata after each calibration event. By reducing the

framerate, many of these frames get lost. We informed the manufacturer that this is relevant information that should be included in the manual.

L130: Define FPA acronym used in figure in caption.

Added to caption: "The temperatures of the focal plane array (FPA) and housing case returned from the sensor are shown in orange and bold black dashed lines."

L137: Change 'asserting' to 'assuming' – this is an assumption that you have made for the final calibration, the ice surface may be 0°C but ice surface temperatures can be colder than 0°C and this should be acknowledged.

Done.

It is also apparent that after your correction, some ice surfaces are now pushing 10°C (frame number ~250 (c)) where the spline interpolation was not super effective, this should be explained and acknowledged too.

Good point. We added: "However, for large ice temperature variation, the spline interpolation may not capture the temperature offset as shown in Fig. 2c (frame ~250)."

L145: State number of thermal and optical images used in this sentence.

We understand that this section needs more information and changed the text accordingly, including the number of images used.

L155: State size of test sight so comparison with the footprint of the reanalysis data is possible.

We added "(~150 m × 350 m)".

L166: Do you mean 'by accounting for the water vapor content…'?

Thanks for pointing out, that this is unclear. We do not account for water vapor in the atmosphere between the surface the and sensor when calculating the LST. We assume water vapor to have a minor influence on the LST measurement and would be compensated by our bulk correction approach. To be clearer we changed the sentence to: "We think, radiation attenuated by water vapor in the atmosphere between sensor and ground would be spatially uniform and thus compensated by our calibration procedure. "

L168: Why/how are they the best classification results?

Good point! By comparison with the optical imagery and according to the algorithms' Out-of-bag (OOB) error, ice surface faces were best discriminated from surrounding debris when the temperature difference was largest, at 15 h. To be clearer on this point we changed the sentence to: "By comparison with the optical imagery and according to the algorithms mean prediction error, we found best classification results when the temperature differences between ice and debris were the largest, at 15 h. Data from this flight were used to classify the thermal imagery."

L169: Add statement to end of sentence along the lines of 'thus data from this time stamp were used to classify the thermal imagery'.

Done.

L172: I'm not sure I understand how ΔS is a rate of change if the right-hand side of Eq. 2 is fluxes in W m-2?

ΔS is the rate of change of heat stored in a layer of debris (see e.g. Brock et al, 2010). To balance surface energy fluxes at sub-daily time intervals, the debris layer must be treated as a volume with variable heat storage. It requires knowledge of the average rate of mean debris temperature change, which we obtain by fitting a sine function to the diurnal LST variation in each pixel of the map and forming the first derivative for each time of flight (described in L220-L225 and examples shown in Fig. 5).

L198: Can you comment on the accuracy of wind speed data taken from reanalysis products and the impact this will have on your subsequent debris thickness estimates (see Schauwecker et al 2015; Stewart et al 2021)?

Good point. There is no doubt that the accuracy of wind speed from reanalysis data is low and not well suited for a small valley glacier with a complex wind regime. We address that point in section 5.1 with our sensitivity test. Figure 12 shows the impact of wind speed (and other parameters) on the debris thickness estimates at the different times of the day by the coverage panel. In short, the wind speed has a strong impact on the sensible heat flux and therefore on b (Eq. 8). In the first half of the day this leads to unphysical solutions of Eq. 7. We discuss that point in detail in section 5.1. Interestingly, in the afternoon hours, wind speed has no big influence on the SEBM-derived results.

L203: State what your definition of 'thin' debris thickness is.

Good point. We added: "(<10 cm)" to the text, based on results shown in Conway and Rasmussen (2000), Nicholson and Benn, 2006; Rounce and McKinney, 2013, which we cite in the text.

L229: Can you please justify why the debris thickness was estimated by solving a quadratic rather than previously documented methods such as that in Rounce and McKinney, 2014? The exclusion of sites for which there is not a real or positive solution to the quadratic equation means that a large proportion of your data is excluded from further analysis – which poses quite a large problem when your study area is already quite small.

In our study, we extended the SEBM by the component ΔS compared to Rounce and McKinney, 2014. Using remotely sensed LST data this has not been done before as most remote sensing products do not provide LST data on sub-daily time intervals. ΔS (the change in heat storage) itself is a function of debris thickness. Consequently, the SEBM now contains G(d) and ΔS(d) which then results in a quadratic equation when solving for debris thickness.

L230: Change testing to training.

Done.

**RESULTS**

L236: I think it is the estimation of debris thickness that changes rather than the relationship between LST and debris thickness, no? i.e. when hotter LSTs are observed, thicker debris will be estimated?

True. We changed the sentence to: "The LST changes over the day in a cyclic manner (early morning cool – afternoon hot – evening cool) and consequently the ability to estimate debris thickness using LST changes accordingly (Fig. 3)."

L240: A key problem for me with this figure is the lack of consistency between the areas of ice and debris in each subplot – theoretically, these areas should remain consistent throughout the day, the time scale of this study is not large enough to observe actual change in the cliff geometries. This then throws into question the accuracy of the data in each time stamp if cliffs are not consistently detected. Can you provide an explanation / justification for this?

Yes, we agree. Based on a similar comment of reviewer 1, we paste here our response:

Thanks for pointing out the inconsistency in ice cliffs – that's an important point! The inconsistency in the bare ice geometry is due to the residual uncertainties of the ice surface temperature. Figure 2 shows that the ice surfaces can vary by several degrees (~6 K). This means that our threshold of 0.5 K in figure 3 does not capture all ice surfaces. We added to the caption of figure 3: "Due to the residual uncertainties of the LST, ice surface geometries appear inconsistent in time."

L243: Can you show a linear regression line and an R2 value on each subplot to support this statement?

Good idea. Done.

L270: Are these LST ranges using raw LST or offset corrected LST?

These are corrected LST.

L295: In caption (or next to the color bar in the figure) state what aspect degrees refer to (i.e. N S E W). Also, is the debris thickness manually measured debris thickness? Make this clear.

Changed to: "… colorized by debris thickness measured in the field." AND "… with lines colorized by terrain aspect with 0/360 ° facing north."

L310: To support this statement, include a histogram of manually measured debris thicknesses for comparison with the modelled debris thicknesses in Fig. 7.

Great idea. We added a histogram to figure 1.

L312: I am concerned about how much data is not valid in Fig7a-d, and I'm not convinced that 'no valid solution' is a sufficient explanation for the lack of data. A surface energy balance model should not be unsolvable. To compare data from different time stamps, the data needs to be (and should be) spatially consistent.

Thanks for pointing that out. Reviewer 1 had similar concerns; therefore, we copy our answer here:

Yes, we agree with the reviewer that these unphysical results are relevant, especially in the morning flights. We also agree that these uncertainties may raise concern for areas with existing solutions. We devoted much of chapter 5.1 to discussing and explaining the cause of this issue. In short, such issues arise specifically when the debris is heating up from morning to mid-day when uncertainties in LST or reanalysis-derived quantities can cause no solution to the equation. In addition, we address the impact of several parameters on the model and the unphysical results (sensitivity test Figure 12).

We expanded the discussion section in 5.1 on these issues in the revised version and changed the sentence in the caption text of Figure 7 to: "White regions show regions where the surface energy balance model has no valid solution for debris thickness due to high uncertainties in the land surface temperature and reanalysis data".

L318: Quantify 'pattern of thin debris predictions'.

Changed to: "Predictions of thicker (>10 cm) debris are primarily found in the afternoon and evening hours (17 h to 22 h) and the pattern of thin debris (<10 cm) predictions, primarily in the central part of the glacier, is relatively consistent in time."

L343: Sentence beginning 'Predicted debris thicknesses…' does not make sense.

Changed to: "The modelled debris thicknesses range varies between 0 cm and 30 cm, but with early flights at 9 h and 11 h lacking predictions greater than 10 cm (Fig. 10a, b)."

**DISCUSSION**

L374: underestimates compared to what?

Changed to: "The SEBM approach mostly underestimates debris thickness at all flight times compared to field measurements."

L406: Figure units! X axis and also mean debris thickness in top right.

Done.

L417-421: My takeaway from this is that 1) wind speed is not modelled well with ERA-5 data, 2) if wind speed is increased to 'realistic' values then the amount of 'valid' debris

thickness pixels decrease significantly? Can you discuss what implications this has in terms of the methodology? I.e. would you recommend that SEBM are not used in conjunction with thermal data to estimate debris thickness?

> Correct, we added a sentence to the discussion: "The high sensitivity of the SEBM approach to uncertainties in LST and the reanalysis data, reduce the suitability to reliably estimate debris thickness."

L434: Can you quantify 'in parts more accurate'?

> To be more precise we changed the sentence to: "Debris thickness predictions using least squares regression of a rational curve yield RMSE values between 6 and 8 cm, similar to the results of the SEBM approach"

L451-452: Westoby et al 2020 do this with optical imagery and a geodetic based debris thickness estimation.

> Good point. We changed the text to: "High resolution studies can, improve our understanding of processes that move and distribute debris on glacier surfaces (Westoby et al., 2020). Spatiotemporal debris thickness estimates using UAV derived thermal images has the potential to serve as a new approach to quantify how debris is mobilised across the surface of the glacier over short times scales."

L455: It would be good to see a debris thickness difference map (i.e. rational curve – sebm debris thickness) to quantify the differences between the two methods as panels c and d visually look very different. Are the differences between the two methods significant on a pixel-by-pixel basis?

> Great idea. We will add a panel showing the difference between the two approaches and a second panel showing a 2d histogram between both model results in the revised version.

L512-520: I would like to see a recommendation of 1) which method is the better predictor of debris thickness, and 2) at which time of day the method appears to be the most accurate. This seems to be missing from the discussion and the paper in general.

> True, we add: "However, as this approach is less sensitive to uncertainties in LST, we recommend the rational curve approach to estimate debris thickness as long as enough debris thickness measurements are available."

Technical corrections:

Ensure LST and not LST's throughout the paper.

> Done.

Rounce and McKinney, 2013 – should this be Rounce and McKinney, 2014, see reference below?

> Thanks, done.

L150: Check section numbers here and throughout (i.e. where they're referenced, such as in L187).

Thanks, done.

L362: Figureb?

Done.

L364: that thin debris should be than thin debris.

Done.